# Biological Properties of SARS-CoV-2 Variants: Epidemiological Impact and Clinical Consequences

**DOI:** 10.3390/vaccines10060919

**Published:** 2022-06-09

**Authors:** Reem Hoteit, Hadi M. Yassine

**Affiliations:** 1Clinical Research Institute, Faculty of Medicine, American University of Beirut, Beirut 110236, Lebanon; rah84@mail.aub.edu; 2Biomedical Research Center and College of Health Sciences-QU Health, Qatar University, Doha 2713, Qatar

**Keywords:** severe acute respiratory syndrome coronavirus 2 (SARS-CoV-2), coronavirus disease 2019 (COVID-19), variants of concern (VOCs), mutations, epidemiological factors, clinical consequences

## Abstract

Severe acute respiratory syndrome coronavirus 2 (SARS-CoV-2) is a virus that belongs to the coronavirus family and is the cause of coronavirus disease 2019 (COVID-19). As of May 2022, it had caused more than 500 million infections and more than 6 million deaths worldwide. Several vaccines have been produced and tested over the last two years. The SARS-CoV-2 virus, on the other hand, has mutated over time, resulting in genetic variation in the population of circulating variants during the COVID-19 pandemic. It has also shown immune-evading characteristics, suggesting that vaccinations against these variants could be potentially ineffective. The purpose of this review article is to investigate the key variants of concern (VOCs) and mutations of the virus driving the current pandemic, as well as to explore the transmission rates of SARS-CoV-2 VOCs in relation to epidemiological factors and to compare the virus’s transmission rate to that of prior coronaviruses. We examined and provided key information on SARS-CoV-2 VOCs in this study, including their transmissibility, infectivity rate, disease severity, affinity for angiotensin-converting enzyme 2 (ACE2) receptors, viral load, reproduction number, vaccination effectiveness, and vaccine breakthrough.

## 1. Introduction

Human coronaviruses (HCoVs) infect a wide range of species, affecting respiratory, gastrointestinal, liver, and nervous systems [1,2]. In humans, they cause mild to severe respiratory infections [3]. They belong to *Coronaviridae* family, order Nidovirales. The coronavirus family is divided into four subgroups: Alpha (α), Beta (β), Gamma (γ), and Delta (δ) [3,4]. The name coronavirus (CoV) is derived from the Latin word corona, which means “crown”. This is due to the virus’s unique structure, which resembles a crown due to surface projections on the viral envelope. Coronaviruses are single-stranded, positive-sense RNA viruses with a diameter ranging from 60 nm to 140 nm and a genome of around 30 kb [5], making them the world’s largest RNA viruses [3]. The genome encodes four major structural proteins: spike (S), small protein (E), matrix (M), and nucleocapsid (N) [6]. The virus’s infectivity and transmissibility in the host are preliminary determined by S protein [7].

Seven types of CoVs have been documented to infect humans, including two *alpha*-CoVs and five *beta-coronaviruses* (β-CoVs). The α-CoVs are HCoV-229E and HCoV-NL63, and the five β-CoVs are HCoV-HKU1; HCoV-OC43; severe acute respiratory syndrome coronavirus (SARS-CoV); Middle East respiratory syndrome coronavirus (MERS-CoV); and the most recent, SARS-CoV-2 [3,4]. SARS-CoV-2 is the etiological agent of COVID-19, and it shares roughly 80% nucleotide identity with SARS-CoV [8,9].

According to a systematic review and meta-analysis conducted by Cevik et al. (2021) [10] the three highly pathogenic human coronaviruses are SARS-CoV-2, SARS-CoV, and MERS-CoV [10]; however, the ongoing epidemic has claimed the lives of a high number of people in comparison to the six coronaviruses that have circulated in the human population [11,12,13]. As of May 2022, the World Health Organization (WHO) reported more than 500 million confirmed cases of COVID-19 and more than 6 million deaths worldwide [14]. The Americas and Europe have reported the most confirmed cases, with more than 100 million each, whereas Africa has experienced the fewest, with more than 11 million reported cases. Furthermore, the Americas are at the top of the pyramid, with nearly 2.5 million deaths, whereas the western Pacific and Africa are at the bottom, with more than 250 thousand reported deaths by April 2022 [14]. Despite being the world’s second-most populous continent, with an estimated 17.2% of the world’s population, Africa accounts for only 5% of overall cases and 3% of death [15].

CoVs are one of the most quickly evolving viruses [16] and have a high rates of mutation and recombination when compared to other RNA viruses, which allows them to adapt to different hosts and spread across species [17]. According to the literature, the sequence diversity of SARS-CoV-2 and its overall evolutionary rate seems to be low [18,19,20]. SARS-CoV-2 encodes an exonuclease (ExoN) that contributes to the genome error repair process [21]. SARS-CoV-2 appears to have a greater mutation rate per site per year than the influenza virus: (1.12–6.25) × 10^−3^ vs. (0.60–2.00) × 10^−6^, respectively [18,22,23]. On the other hand, MERS-CoV has a mutation rate of 1.12 × 10^3^ substitutions per site per year in the whole genome [24], whereas HCoV-OC43 and HCoV-229E have a mutation rate of around 3~6 × 10^4^ substitutions per site per year [25,26].

Despite this, viral mutations do occur, and their frequency has increased as a result of natural selection of favorable mutations, random genetic drift, recombination, or epidemiological factors. A mutation is defined as any change in a virus’s genetic sequence that differs from the normal sequence, such as a substitution, deletion, or addition [27]. SARS-CoV-2 naturally mutates as it spreads around the world, resulting in new variants that are either more or less infectious/pathogenic, depending on the modified composition [27]. New variants are categorized based on their potential impact on transmissibility, severity, and/or immune evasion, all of which are likely to influence the epidemiological situation (number of cases, hospitalizations, intensive care unit admissions, etc.) [4,28,29,30]. Since January 2020, the WHO has been monitoring and assessing the evolution of SARS-CoV-2 in collaboration with partners, expert networks, national authorities, institutions, and researchers. The identification of specific variants of interest (VOIs) and variants of concern (VOCs) in late 2020 was prompted by the emergence of variants that posed an increased risk to global public health (WHO, 2021) [14]. The variants are classified into three categories: variants of interest (VOI) if preliminary evidence indicate a potential impact, variants of concern (VOCs) if the impact is known to be significant, and variants under monitoring (VOM) [31] (Figure 1 and Figure 2). Current VOCs are Alpha, Beta, Gamma, Delta, and Omicron. According to Parums (2021) [32], there has been a lack of a consistent scheme for designating SARS-CoV-2 variants of concern (VOCs) and variants of interest (VOIs) throughout the last 18 months. The global initiative on sharing all influenza data (GISAID), Nextstrain, and Pango are three scientific nomenclature systems that have been used to identify and track SARS-CoV-2 genotypes [33]. On 31 May 2021, the World Health Organization’s (WHO) Virus Evolution Working Group announced its suggestions for naming SARS-CoV-2 VOCs and VOIs [32]. The WHO assigned simple labels to key SARS-CoV-2 variants based on Greek alphabet letters. These labels were chosen following extensive consideration of a variety of proposed naming systems [31].

COVID-19 transmission is exponential, as evidenced by the number of new cases, admissions, and deaths. The basic reproduction number (R_0_) and the real-time effective reproduction number (R_t_) are two measures used to assess pandemic trends and quantify a pathogen’s epidemic potential [34,35,36,37]. R_0_ or R-naught represents the virus’s initial reproduction number at the start of the pandemic, and it is used to measure the transmission potential of a disease. R_0_ represents the average number of secondary cases produced by a typical case of infection in a population where everyone is susceptible [37]. It is a constant value that describes what happens during a pandemic when no public health policies are implemented and cannot account for the epidemic’s time-varying nature [38]. Several factors influence the basic reproduction number: the rate of contacts in the host population, the possibility of infection spreading during contact, and the duration of infectiousness [37].

R_0_ is calculated as R_0_ = $\frac{\beta }{\gamma }$, where γ = 1/average infectious period and β is the transmission rate or the number of contacts of an infective case in a defined time [39].

For example, if R_0_ for COVID-19 is equal to 4, it suggests that a single infected individual has the ability to spread the virus to four persons.

If R_0_ < 1, each current infection creates fewer than one new infection. In such a situation, the disease will gradually deteriorate and eventually disappear. If R_0_ = 1, each current infection produces a new infection. However, in such a case, there will be no outbreak because the disease will continue to exist and remain stable. If R_0_ > 1, each current infection produces more than one new infection. Thus, the disease will spread from person to person, leading to an outbreak or epidemic [40].

On the other hand, R_t_ is the effective reproduction number and is defined as the average number of secondary infections generated by each infection at any given time in a population made up of both susceptible and non-susceptible hosts. In other words, “it is the expected number of new infections caused by an infectious individual in a population where some individuals are immune” [41]. R_t_ has proven to be one of the most effective tools for monitoring and tracking the epidemic (by assessing whether it is growing, shrinking, or holding), as well as informing escalation and easing of restrictions [41,42,43,44]

The effective reproduction number can be calculated by multiplying the basic reproduction number by the fraction of the host population that is susceptible (x) [37].
R = R_0_x

R_t_ ≥ 1 denotes increased growth, whereas R_t_ < 1 denotes infection disappearance.

If the population is half immune, the effective reproduction number will be 4 × 0.5 = 2. This means that on average, a single case of COVID would infect two people.

R_t_ can also be calculated using another method [45].
R_t_ = c × p × D × S

R_t_ is presented in terms of the population’s social contact rate (c), the likelihood of the virus being transmitted in one contact between an infectious person and a susceptible individual (p), the duration of infectiousness (D), and the proportion of the population that is still infectious (S).

c represents how frequently people meet; p is the transmission likelihood of the infection per contact, D is the duration over which an infected individual can pass the infection to another individual, and S is proportion of the population not yet immune.

On the other hand, herd immunity occurs when a considerable proportion of the population has been vaccinated, resulting in the protection of unvaccinated individuals. Therefore, the greater the number of immune persons in a population, the less likely a susceptible person is to become infected [37].

The herd immunity threshold refers to the proportion of a population that must be immune for an infectious illness to be stable in that community.
HIT = R_0_ − 1/R_0_

If this is achieved, for example, by vaccination, each case will result in only one new case (R = 1), and the infection will become stable in the population. However, if the herd immunity threshold is exceeded, R becomes smaller than 1, and the number of infection cases declines [37].

Moreover, the incubation period, often known as the time between initial exposure and disease onset, is a critical metric for determining the transmission of infectious diseases and determining quarantine measures. The mean incubation period, for example, is frequently used to compute the reproduction number, whereas the maximum incubation period is frequently employed to establish quarantine duration [46]. COVID-19 has a wide range of incubation times, ranging from 2.87 days [47] to 17.6 days [48].

Long COVID, also known as chronic or long-haul COVID, is a set of more than 50 symptoms that can linger weeks or months after a COVID-19 infection, according to the Centers for Disease Control and Prevention (CDC) [49]. According to research, even minor occurrences of COVID-19 can result in a long COVID [50]. “Long COVID can develop no matter what viral variants emerge”, said Dr. Anthony Fauci. Preliminary research indicated no difference between Delta, Beta, and Omicron [50]. However, according to new research, long COVID symptoms may vary depending on which SARS-CoV-2 variant caused the infection [51]. For example, when the Alpha variant was dominant, the prevalence of myalgia (10%), dyspnea (42%), brain fog/mental confusion (17%), and anxiety/depression (13%) rose considerably compared to the wild-type variant, whereas anosmia (2%), dysgeusia (4%), and impaired hearing (1%) were less common. However, when the wild-type variant was prevalent, fatigue (37%), sleeplessness (16%), dysgeusia (11%), and hearing loss (5%) were all more common than when the Alpha variant was dominant, followed by brain fog (10%), myalgia (4%), and anxiety/depression (6%) [51]. Long-term COVID symptoms and other SARS-CoV-2 variants should be investigated further. In addition, extended COVID has been shown to be more detrimental in severe cases of inflammation. According to a previous study, as many as 30% of patients acquire long COVID [52].

According to the literature, human genetic variants can influence the severity of infectious diseases, including SARS-CoV-2 infection [53,54]. Genetic factors have a wide variety of impacts, from uncommon, high-impact mutations that can make a massive difference between mild symptoms and life-threatening sickness to more frequent genetic variants that only have a moderate impact on symptom severity [54,55]. It is of note that researchers discovered 13 genes (or loci) in the human genome that influence COVID-19 susceptibility and severity [56]. Furthermore, the researchers looked for ‘candidate genes’ and identified more than 40 potential genes, many of which have be linked to immune function or have lung-related functions [56].

On the other hand, clinical trials and real-world evidence suggest that vaccines are highly successful in preventing hospitalization and mortality linked to SARS-CoV-2 infection [57]. Vaccines have been produced and deployed quickly to combat the global spread of COVID-19, including seven vaccines approved by the World Health Organization (WHO) for emergency use (EUL) [58,59]. The most commonly utilized and first approved mRNA vaccines, Pfizer-BioNTech (BNT162b2) and Moderna (mRNA-1273), were rolled out worldwide in December 2020 and played an essential role in slowing the spread of SARS-COV-2 [60]. Research shows that COVID-19 vaccines provide adequate protection against SARS-COV-2 infection and significantly reduce the risk of severe illness [61]. SARS-COV-2 infection is best prevented with mRNA vaccinations. The COVID-19 vaccines were found to protect against SARS-COV-2 infection in the majority of investigations, with efficacy improving after the second and booster dosages. Virus vector vaccinations, on the other hand, provided at least 4% less protection compared to mRNA vaccines [61]. In terms of vaccination safety, mRNA vaccines had a higher correlation with major adverse events than viral vector and inactivated vaccines, although there is no convincing proof that COVID-19 vaccines directly cause serious adverse events. With the rapid evolution of SARS-CoV-2 globally, a long-term surveillance mechanism for the safety and efficacy of COVID-19 vaccines is required, as well as more research.

Moreover, several factors can speed up or slow down the spread of COVID-19, including the mode of transmission, the role of asymptomatic infected people, the mode of replication in the upper respiratory tract vs. lower respiratory tract, herd immunity, urban or rural settings, population density, age, weather conditions, and vaccinations. The purpose of this review article is to look into the key variants of concern (VOCs) and mutations of the virus causing the current pandemic, as well as to investigate SARS-CoV-2 VOC transmission rates in relation to epidemiological factors and to compare the virus’s transmission rate to that of previous coronaviruses. The findings related to variants of concern are summarized in Appendix A (Table A1).

### Virus Transmission between Sick and Asymptomatic Individuals

The virus spreads from person to person via respiratory droplets (within six feet), regardless of clinical illness (sick or asymptomatic) [62]. Aerosol transmission and, possibly, contact with fomites are also possibilities, although these are not thought to be the most likely routes [63]. Disease transmission is also influenced by the variant strain, as well as environmental factors [64,65]. The infectivity of SARS-CoV-2 is partially dependent on the viral spike protein binding to the angiotensin-converting enzyme 2 (ACE2) receptor, with ACE2 receptor cleavage by a type 2 transmembrane serine protease (TMPRSS2) required to activate the spike protein [66]. SARS-CoV-2 enters endosomes after attaching to the cell surface ACE2 receptor via spike glycoprotein. The S protein is made up of two domains: S1 and S2. The S1 domain contains a signal peptide, an N-terminal domain, and the receptor-binding domain (RBD, 438–506 a.a.), which aids in the identification of human ACE2 receptors [67]. Thakur et al., 2021 showed that the RBD domain often resonates between standing and lying-down positions. Surprisingly, the RBD domain in SARS-CoV-2 persists in a lying-down orientation, facilitating immune evasion and high infectivity [67]. SARS-CoV-2 VOCs refers to viral variants having mutations in their spike protein receptor-binding domain (RBD) that drastically boost binding affinity in the RBD-hACE2 complex while also generating rapid proliferation in human populations [68]. SARS-CoV-2 mutations are more likely to develop when viral replication is increased [67]. The evolution of SARS-CoV-2, starting at the end of 2020, was distinguished by the introduction of ‘variants of concern’ or alterations in viral features, such as disease transmission and antigenicity, most likely due to changes in the human species’ immunological composition [66].

## 2. Variants

### 2.1. D614G Variant

Originally from China, the D614G variant of SARS-CoV-2 made its way to Europe in January 2020 and quickly became the world’s most prevalent strain in just three months. The D614G mutation, the first to occur in the S glycoprotein outside the RBD region, was first detected in Germany in January 2020, and by June 2020, it had become the prevalent mutation in all circulating strains globally [69]. The D614G variant (also known as G614), which resulted from a D-to-G amino acid shift produced by a single-nucleotide mutation at position 1841 of the Sgen in the Wuhan reference strain, was the primary circulating variant of SARS-CoV-2 until the beginning of 2021 [70]. The switch from D614 to G614 happened asynchronously in different parts of the world, starting in Europe, then North America, Oceania, and Asia [69]. G614 had become the pandemic’s prevalent type by early April 2020. Because the mutation is present in all documented VOCs, it is considered very important. It is worth noting that the major mutation D614G is linked to the new variants’ greater transmissibility [71].

#### 2.1.1. Transmissibility

D614G enhances SARS-CoV-2 replication in cultured human nasal epithelium and was found to promote transmission in a hamster model, according to a study published in Science by Hou et al. (2020) [72]. The D614G variant improves virus replication in human lung epithelial cells and primary human airway tissues by enhancing the infectivity and stability of virions, according to experimental findings. Infectious titers in nasal washes and the trachea are higher in hamsters infected with the D614G variant, indicating that the G614G mutation plays a role in enhanced transmissibility [73].

#### 2.1.2. Infectivity Rate

The D614G mutation has been linked to increased virus infectivity and transmissibility [69,74]. The binding affinities of SARS-CoV-2 Spike D614 and G614 variants for hACE2 are comparable, a K_dis_2 being of 2.01 × 10^−2^ and 2.06 × 10^−2^, respectively [74]. According to Ogawa et al., 2020, the D614G variant spike exhibits >1/2 log_10_-enhanced infectivity in human cells expressing the human ACE2 protein as the viral receptor [75]. A study conducted by Korber et al. (2020) [69] on the G614 frequencies in the five continents and including 17 countries revealed that G614-bearing viruses have much greater infectious titers (2.6- to 9.3-fold increase) than D614-bearing viruses, which was validated in multiple cell types [69].

#### 2.1.3. Disease Severity

Clinical investigations yielded no evidence that patients infected with the spike 614G variant have a greater COVID-19 mortality or clinical severity, although 614G is linked to a higher viral load and a younger patient age [69,71]. Another study conducted by Pandey et al. (2021) suggested that there is an association between severe disease and phylogenetic clade 20C [76].

#### 2.1.4. Affinity to Angiotensin-Converting Enzyme 2 (ACE2) Receptors

Despite the fact that D614G is located outside of the RBD at the C terminus of the S1 domain, it was thought that this single mutation would improve S-ACE2 interaction by promoting the open conformation of RBD [77], which could plausibly increase S-ACE2 binding. The increased infectivity of the D614G mutation could also be due to a change in the binding affinity between S and ACE2 [77,78]. Additionally, the D614G mutation promotes cell entrance by acquiring a greater affinity for ACE2 while preserving neutralizing susceptibility, according to Ozone (2021) [79].

#### 2.1.5. Viral Load

Researchers from the University of Sheffield and the University of Washington discovered that COVID-19 patients with the G614 variant exhibit a threefold increase in viral RNA as tested by quantitative PCR (qPCR) [69,74]. In human patients, G614 was also linked to higher levels of viral nucleic acid in the upper respiratory tract [69]. It has been indicated that the G614 variant is connected with potentially higher viral loads (Ct value between 20 and 30) but not with disease severity based on Ct-value analysis [69]. Ct is utilized as a proxy for relative viral loads; lower Ct values imply higher viral loads [80]. Another study that used the 2019-nCoV N1 real-time qPCR assay to analyze real-time quantitative viral load in a subset of 31 614D and 290 614G samples reported a significant difference in 614G association with higher viral load (*p* = 0.0151) [71]. In addition, a study conducted in India between March and June 2020 found that in comparison to isolates with the D614 mutation (Ct value between 17 and 23), isolates with the G614 mutation have lower Ct values (18–22) [81]. Further research conducted in the US found a link between the D614G mutation and Ct values, implying that the variant is linked to a higher viral load and that the D614G mutation makes the virus more infectious [82]. Researchers discovered that the G form is extremely infectious and is linked to higher viral loads in the upper part of the respiratory tract through in vitro investigations [83]. Similarly, studies in a Chicago cohort found that G614-expressing strains were linked to increased viral loads in the airways but not to worse clinical outcomes [84].

#### 2.1.6. Reproduction Number (R_0_/R_t_)

According to a statement issued by the World Health Organization (WHO) on the outbreak of SARS-CoV-2 on 23 January 2020, the basic reproduction number (R_0_) of COVID-19 was initially predicted to be between 1.4 and 2.5 for D614 and G614 variants [85]. During the early epidemic phase, in three nations (Australia, the UK, and USA), the R_0_ for D614 was similar (from 1.56 for USA to 1.73 for Australia) [86]. Liu et al. examined 12 studies that evaluated the R_0_ of COVID-19 between 1 January and 7 February 2020 and found a range of values between 1.5 and 6.68 [86]. The authors assessed the mean and median R_0_ values estimated in the 12 studies and found that COVID-19 had a final mean and median R_0_ value of 3.28 and 2.79, respectively, with an interquartile range (IQR) of 1.16. However, a considerably differing R_0_ was found among countries (ranging from 1.82 in the United Kingdom to 3.87 in the United States), demonstrating that the spatial transmission of G614 varied more than that of D614 (Wang 2020) [87]. A study conducted in three different regions in China using data up to April 2020 revealed varying R_0_ values, with R_0_ for regions 1, 2, and 3 of 1.94, 1.62, and 2.19, respectively [88]. According to data obtained in Bangladesh between 8 March and 31 July 2020, the overall basic reproduction number fluctuated with time, with the highest R_0_ in the 4th week (29 March–4 April) (R_0_ = 4.86) and the lowest R_0_ in the 20th week (19–25 July) (R_0_ = 1.09) [89].

#### 2.1.7. Vaccine Effectiveness and Vaccine Breakthrough

The level of vaccination efficacy needed to achieve a societal level of protection is a tricky issue to answer. According to computational modeling and simulation tests, if vaccination coverage is 100%, a minimum of 60% vaccine efficacy is required to halt an ongoing epidemic [90]. Given that 100% vaccine coverage is highly improbable due to a variety of factors, such as vaccine hesitancy and shortages in many regions of the world, the necessary efficacy rises even more. The same study discussed that for a vaccination coverage of 75%, to end a current pandemic, 80% efficacy is required. According to another study, an efficacy of 50% or more could significantly lower the prevalence of COVID-19 in vaccinated people and provide meaningful herd immunity [91].

According to a study estimating the transmission advantage of the D614G mutant strain of SARS-CoV-2 from December 2019 to June 2020, the D614G mutation would raise the herd immunity threshold from 50% to 62% (i.e., 12% excess) if R_0_,D614 = 2 and from 67% to 75% (i.e., 8% excess) if R_0_,D614 = 3 [92].

In summary, the D614G mutation is considered very important because it is present in all identified VOCs. Increased virus infectivity and transmissibility have been attributed to the D614G mutation. There is no evidence that people infected with the spike 614G variant experience higher COVID-19 mortality or severity. The D614G mutation enhances cell entry by increasing ACE2 binding while maintaining neutralization susceptibility. Researchers discovered a relationship between the D614G mutation and Ct values, implying that the variant is associated with a larger viral load and that the D614G mutation makes the virus more infectious but not leading to worse clinical outcomes. Depending on the region and time, the reproduction number fluctuates from one to four.

### 2.2. Alpha Variant

The Alpha strain (B.1.1.7) was first detected in the United Kingdom in November of 2020, and infections spiked in December of that year [93]. This variant is also known by the names 20I/501Y.V1 in the Nextstrain database and GR/501Y.V1 in the GISAID database [94]. There are 23 mutations in total. E484K, S494P, and N501Y are all important mutations in the “S” receptor-binding domain (RBD) [94]. In addition to the RBD-area mutations, other notable mutations in S glycoprotein include 69del, 70del, D614G, 144del, A570D, S982A, P681H, D1118H, T716I, and K1191N [95]. Researchers from the United States and the United Kingdom collaborated to learn more about how Alpha attacks the human body and discovered that the mutations that allow it to thrive are not limited to those centered on the spike protein [96]. They also discovered that the Alpha variant increased the production of N and Orf9b protein, which could aid in suppressing the mechanism by which infected cells communicate with the immune system [96]. Three viral proteins that are known to enable COVID-19 to escape the immune response were found in abundance within Alpha-infected cells. Orf9b, in particular, did so by inhibiting RIG-I-MAVS in cells, which normally turns on genes that promote the immune system reaction.

#### 2.2.1. Transmissibility

Increased transmissibility of this variant is linked to the major mutation D614G [97]. The UK experienced the highest death rate in January 2021. The B.1.1.7 variant then spread to 30 other nations, including the United States, where it was discovered between December 2020 and January 2021 (Chakraborty 2021) [94]. Moreover, by February 2021, it had spread to more than 40 states in the US [98]. According to the Global Virus Network, the variant has spread to at least 114 countries [99]. Whereas this variant was once considered to account for roughly 70% more transmissible than the original (wild-type) SARS-CoV-2 coronavirus, new research suggests that it is about 30–40% more transmissible [100]. Similarly, Yang and Shaman estimated that B.1.1.7 is 46.6% (95% CI: 32.3–54.6%) more transmissible but exhibits nominal immune escape from protection caused by prior wild-type infection using a model-inference system they developed to estimate epidemiological properties of new SARS-CoV-2 variants of concern [101]. Additionally, this variant was estimated to be 36–55% more transmissible than other circulating lineages in Denmark around the same period [102]. The same study conducted by Lyngse et al. (2021) revealed that primary cases infected with B.1.1.7 exhibited a 1.5–1.7 times higher transmissibility than primary cases infected with previous lineages [102].

The proportion of SARS-CoV-2 variants (501Y, B.1.1.7) was predicted to be 67% in Geneva and 35% in Zurich on 5 February 2021, according to Althaus et al. (2021), who also projected that the increase in transmissibility would be slightly higher than 50% [103].

Another study conducted by Davies et al. (2021) in the UK revealed that this variant is 43 to 90% (95% credible interval (CrI): 38–130%) more transmissible than the preceding lineages [104]. A similar result was reported in another study conducted between October and November 2020 in the UK by Leung et al. (2021), which revealed 75% higher transmission [63]. Based on 94,934 SARS-CoV-2 sequencing data and COVID-19 surveillance data collected between 1 August to 31 December 2020, Zhao et al. estimated that B.1.1.7 is 52% (95% CI: 46, 58) more transmissible than the wild type in the UK, according to public databases released by GISAID and the World Health Organization [105,106]. Brown et al. (2021) demonstrated increased transmissibility of B.1.1.7 in Ontario, Canada (OR adjusted = 1.49, 95% CI: 1.36–1.64) [107].

In a study conducted across the United States of America, Washington et al. (2021) indicated that this variant had an increased transmissibility of 40–50%, with 29–37% in California and 38–49% in Florida [108].

#### 2.2.2. Infectivity Rate

According to Leung et al. (2021) the percentage of persons infected with the Alpha lineage in the United Kingdom increased from 0.1% in early October to 49.7% in late November 2020 [63]. In December 2020, the England Public Health Authority reported a 3.7-fold rise in COVID-19 cases linked to this new strain [109].

According to a study conducted by Graham et al. (2021) possible reinfections were found in 249 (0.7% (95% CI 0.6–0.8) of 36,509 app users who reported a positive swab test before 1 October 2020; however, there was no evidence that the prevalence of reinfections was higher for the B.1.1.7 variant than for pre-existing variants over the same time period [110]. However, the same study found that reinfection was more tightly connected to an overall regional rise in cases than to an increase in the proportion of infections caused by the B.1.1.7 variety [110].

In Qatar, the efficacy of natural infection against reinfection for B.1.1.7 was estimated at 97.5% (95% CI: 95.7% to 98.6%) among persons with a prior PCR-confirmed infection and 97.0% (95%CI: 92.5% to 98.7%) among antibody-positive persons [45].

#### 2.2.3. Disease Severity

Studies conducted in different countries, including the US, Germany, Spain, Denmark, and the United Kingdom, found that the B.1.1.7 lineage is more likely than the original virus to send infected patients to the hospital and is also more deadly [111,112,113]. After adjusting for age, sex, deprivation, ethnicity, region, and week of diagnosis, in a study conducted in England, Nyberg et al. (2021) discovered that patients infected with the B.1.1.7 variant had a 1.52 (1.47 to 1.57) times higher risk of hospital admission within 14 days after a positive test than those infected with wild-type variants [114]. A study conducted in Madrid, Spain, found that patients with the B.1.1.7 lineage were twice as likely to be admitted to the ICU (OR 2.11 95 CI% = 1.55–2.87) [112]. Bager et al. (2021) discovered that infection with lineage B.1.1.7 was associated with a higher risk of hospitalization than infection with previous lineages, with an adjusted OR of 1.64 (95% CI, 1.32–2.04) based on 128 admissions after B.1.1.7 infection and 1107 admissions after infection with other lineages in a Danish study [115]. Furthermore, over a 12-day period in January, the number of patients with B.1.1.7 grew rapidly at a hospital and cardiac center in Lebanon, and the proportion of COVID-19 cases with “spike gene target failure” or “spike gene drop out” (SGTF increased from 16 to 60% [116]; SGTF refers to positive test with a non-detectable S gene Ct value  and Ct ≤  30 for N and ORF1ab targets [117]. A Swedish research group revealed that Alpha-positive persons had considerably greater rates of hospitalization (2.6% vs. 1.2%) and severe illness than negative individuals, although the numbers were too small to assess differences in severity rates among hospitalized individuals [118].

According to preliminary data collected in the United Kingdom, B.1.1.7 infection can result in a 30% to 50% greater mortality rate [119]. Grint et al. (2021) found that Alpha was associated with a 73% higher risk of all-cause death (adjusted hazard ratio aHR: 1.73 (95% CI 1.41–2.13; *p* < 0.001)) and a 62% higher risk of hospital admission (aHR: 1.62 (95% CI 1.48–1.78; *p* < 0.0001)) compared to wild-type virus, accounting for individual-level demographics and comorbidities, as well as regional variation in infection incidence [120]. In comparison to previously circulating variants, the mortality hazard ratio associated with infection with “B.1.1.7” (S-gene-negative) was 1.64 (95% CI: 1.32 to 2.04) in the population in a large study from the UK [121]. After accounting for SGTF misclassification and adjusting for age, sex, ethnicity, deprivation level, care home location, region, and diagnosis date, Davies et al. reported a 35% increase in the risk of mortality within 28 days after diagnosis with B.1.1.7 [122]. On the other hand, Frampton et al. (2021) observed no association between SARS-CoV-2 lineage (B.1.1.7 vs. non-B.1.1.7) and severe sickness or mortality result (unadjusted PR 0.99 [95% CI 0.68–1.43]; adjusted PR 0.99 [0.67–1.47]) [123]. Findings from another study conducted in Spain found no association between the probability of death and the Alpha variant [112].

#### 2.2.4. Affinity to Angiotensin-Converting Enzyme 2 (ACE2) Receptors

The N501Y mutation may boost infectivity by enhancing spike protein binding to ACE2 receptors, according to some research. The N501Y mutation may improve affinity for host cells by enhancing conformational stability, according to Islam et al., when compared to the Wuhan reference strain [89]. Liu and his research team discovered that the mutated Y501 protein exhibited a 10-fold greater binding affinity for ACE2 [124]. Molecular dynamic simulations further suggest the potential of the N501Y mutation having a higher affinity for ACE2 receptors [125,126]. The B.1.1.7 RBD bound ACE2 with a 1.98-fold higher affinity than the SCoV2 RBD [127].

#### 2.2.5. Viral Load

Higher viral shedding and transmissibility may be linked to increased viral load in respiratory specimens [128]. As part of a test and contact-tracing operation, Kidd et al. examined COVID-19 test data received from a laboratory between 25 October and 25 November 2020. The median Ct values for the ORF gene target (22.30 vs. 18.16; *p* < 0.0001) and N-gene target (23.16 vs. 19.39; *p* < 0.0001) were considerably lower in SGTF samples compared to non-SGTF samples. According to Ct values, virus loads in SGTF samples can be 10^4^ times higher than those in non-SGTF samples [129]. Frampton et al. (2021) conducted a study in the United Kingdom and discovered that the viral load was higher in B.1.1.7 samples than in non-B.1.1.7 samples, as determined by cycle threshold value (mean 28.8, SD 4.7 vs. 32.0, 4.8) [123]. In contrast, another study conducted by Walker et al. found that viral loads in 3531 B.1.1.7 samples were not significantly different from viral loads in 8545 non-B.1.1.7 samples and that Ct levels in B.1.1.7 samples dropped significantly from around 30 in mid-November 2020 to around 20 in January 2021, whereas non-B.1.1.7 samples stayed around 22–27 [130].

#### 2.2.6. Reproduction Number (R_0_/R_t_)

Early in the COVID-19 pandemic, the reproduction number in Italy, Spain, France, and Germany was greater than two, resulting in widespread epidemics in these countries [38]. According to research conducted in the United Kingdom, the official total effective reproduction number was primarily between 0.8 and 1.1 on 15 May 2021 [131]. Vöhringer et al. reported an average R_t_ of 1.25 for B.1.1.7 during the lockdown in England in a modeling analysis compared to 0.85 for other circulating lineages [132].

During the time period between 28 September to 27 December 2020, Graham et al. found that in the UK, the R_t_ of B.1.1.7 increased by a factor of 1.35 (95% CI 1.02–1.69) when compared to pre-existing variants [110]. R_t_, on the other hand, dropped below 1 during regional and national lockdowns, even in areas where the B.1.1.7 variant was prevalent. A study conducted by Davies et al. in England estimated R_0_ to be between 1 and 2 in November 2020 [104]. Another study examining data from Lombardia area in Italy during the spread of the B.1.1.7 variant showed an R_t_ = 1.4 in February 2021, which continued to decreased, reaching 1 within a month, with a value 0.8 in April 2021 [133]. Additionally, the R_0_ fluctuated between 0.8 and 1.5 between 1 January 2021 and 26 March 2021 in the Czech Republic [134]. Early data collected from the UK showed that the reproduction number was estimated to be 1.57-fold higher in the B.1.1.7 group compared to the non-VOC group [135]. Furthermore, the R_t_ increased to more than 2 in India between January 2021 to April 2021 during the circulation of the Alpha variant [136]. In Canada, a study conducted in the Greater Toronto Area revealed that the relative reproduction number was 1.44 (95% CI: 1.03, 1.99) between 16 December 2020and 3 February 2021and during the spread of the B.1.1.7 variant [137].

A study conducted in Qatar by Abu Raddad et al. revealed that R_t_ averaged 0.97 from 1 August 2020 to 17 January 2021, whereas it jumped above 1 between 18 January 2021 and 3 March 2021 [45], reaching 1.6 on 24 January 2021.

R_t_ was calculated during weeks 45–55 of the pandemic (1 November 2020 to 16 January 2021) in England. During this time, estimated mean R_t_ ratio of the VOC strains was 1.79 (95% CI: 1.22–2.49). The multiplicative advantage in R_t_ for the VOC dropped across the time window studied, reaching approximately 1.5 in week 55 [97]. The same study conducted by Volz 2021 showed that the R_t_ dropped between late December 2020 and early January 2021, which could be associated with increased social distancing, school closures, and the third England lockdown that followed [97].

Between weeks 43 and 51 (8 weeks), Grabowski et al. found that the R_t_ for the B.1.1.7 over 20E.EU1 was between 1.83 and 2.18 (95% CI: 1.71–2.40) in the UK. The R_t_ for B.1.1.7 of other strains was calculated to be 2.03–2.47 (95% CI: 1.89–2.77), with the lower bound being the estimate for weeks 44–51 and the upper boundaries for weeks 43–47 in the United Kingdom [138].

Moreover, based on an analysis of 15 studies, the effective reproduction number, R_t_, was found to vary from 1.1 to 2.8 [106].

#### 2.2.7. Vaccine Effectiveness and Vaccine Breakthrough

There is no evidence in the literature that B.1.1.7 has a major impact on vaccine efficacy. Preliminary results from a phase 3 trial of the Novavax NVX-CoV2373 COVID-19 vaccine in the UK with more than 15,000 participants aged 18–84 years showed an efficacy of 89.3% (95% CI: 75.2–95.4%) in reducing symptomatic COVID-19 infection (confirmed by PCR). Of the 62 COVID-19 cases studied, 32 were found to be caused by B.1.1.7, 24 by non-B.1.1.7, and 6 by unknown variant, suggesting a vaccine efficacy of 85.6% against B.1.1.7 compared to 95.6% against non-B.1.1.7 strains [139]. Chemaitelly et al. (2021) used a matched case-control study strategy to test the efficiency of Moderna’s vaccination against SARS-CoV-2 variants of concern in Qatar. Effectiveness against B.1.1.7 infection (Alpha) was 88% 14 days after the first dose and 100% ≥ 14 days after the second dose [140]. The Novavax vaccine was found to be 95.6% effective against the original form and 85.6% effective against B.1.1.7 [141]. For maximum efficacy in places where the B.1.1.7 variety is common, it has been recommended that the second dose be given as soon as feasible [142].

Other studies reported no significant impact and no loss of neutralization activity in sera of individuals who received two 100 μg doses of vaccine, using pseudo virus with B.1.1.7 mutations [143,144]. Additionally, serum (n = 14) produced by two doses of the Moderna vaccine resulted in a small but significant reduction in neutralization titers (IC50), according to Wang, Schmidt, et al. When compared to the COVID-19 B.1.1.7 (501Y.V1) variant of concern, the N501Y mutation resulted in a 1- to 3-fold (*p* = 0.0002) reduction in neutralizing activity against pseudovirus [145].

Xie et al. and Wang et al. reported a small but significant reduction in neutralization titers (IC50) in sera elicited by two 30 μg-doses of the Pfizer-BioNTech vaccine (BNT162B2 mRNA) [145,146]. Furthermore, Collier et al. found a modest reduction in neutralizing titers in 20/29 vaccines after one dose of vaccine (mean reduction = 3.2-fold, standard deviation = 5.7-fold) and two doses of vaccine (mean reduction = 1.9-fold, standard deviation = 0.9-fold) [147]. The neutralizing efficacy of sera from 40 participants vaccinated with the BioNTech-Pfizer mRNA vaccine, BNT162b2, against a pseudovirus containing the Wuhan reference strain or the lineage B.1.1.7 spike protein was compared by Muik et al. Variant B.1.1.7 exhibited only a slight reduction in neutralization susceptibility (1.5-fold average) when exposed to convalescent sera [148]. Additionally, based on the WHO reports, there was a reduction in neutralization by less than 2-fold for both BNT162B2 and mRNA-1273 vaccines, 5 to <10-fold for the AstraZeneca-ChAdOx1 vaccine and between 2 and <5-fold for the Johnson & Johnson vaccine (Ad26.COV2.S) [149]

Vaccine breakthrough SARS-CoV-2 infection was observed in 3720 healthcare professionals who had received two doses of BNT162b2 in Italy over the study period (January–May 2021) [150]. SARS-CoV-2 infection was reported in 33 people, with a cumulative incidence of 0.93% after 100 days. When compared to a non-vaccinated control group from the same institution, where SARS-CoV-2 infection occurred in 20/346 subjects, vaccine protection against Alpha-variant infection was 83% (95% CI: 58–93%) in the overall population and 93% (95% CI: 69–99%) in SARS-CoV-2-experienced subjects (100-day cumulative incidence: 5.78%) [150]. In addition, all vaccination breakout infections were asymptomatic or symptomatic, with rhinitis being the most common symptom.

Data from a large healthcare system in the greater New York City area showed that vaccine efficacy is maintained against the Alpha variant more than 14 days after the last dose of vaccine with BNT162b2 (Pfizer/BioNTech), mRNA-1273 (Moderna), or JNJ-78436735 (Janssen). In comparison to a large number of SARS-CoV-2 infections among unvaccinated people, the recorded breakthrough cases (n = 76) accounted for about 1% of total infections between February and April 2021 [151].

In summary, the first VOC is the Alpha variant. The major mutation D614G is linked to its higher transmissibility. In COVID-19 cases, researchers discovered a 3.7-fold increase in infectivity, a 43–90% increase in transmissibility, increased risks of hospitalization, and a probable increase in severity and fatality. High viral loads were detected, with a mean Ct value of 28. Depending on the area and time, the number of reproductions ranged from one to five. Pfizer and Moderna vaccines, as well as AstraZeneca and Johnson & Johnson vaccines, were shown to have reduced neutralization.

### 2.3. Beta Variant

The variant formerly known as the ‘South Africa variant’ is now known as the Beta variant (B.1.351) or 20H/501Y.V2 in the Nextstrain database and GH/501Y.V2 in the GISAID database [94]. The Beta variant of SARS-CoV-2 was detected in South Africa (Eastern Cape region) in December 2020 and quickly became the dominant form, representing more than 95% of cases in the country [152]. Subsequently, the variant was transmitted to numerous other countries. The variant was discovered in India, imported through a traveler, according to Yadav (2021) [153]. In March 2021, a rapid antigen test in Germany detected the variant [154]. Moreover, the variant was discovered in Maryland, USA, by a clinical laboratory in January–February 2021 [155]. However, the Beta form never caused a wave of infections in the United States as it did in South Africa.

The genome of this variant has accumulated a number of mutations (~21) in comparison to the Wuhan strain. One deletion mutation and twelve nonsynonymous mutations stand out. K417N, E484K, and N501Y mutations exist in the RBD area [156]. S-glycoprotein mutations include D80A, D215G, 241del, 242del, 243del, D614G, and A701V, in addition to the RBD [94].

#### 2.3.1. Transmissibility

In terms of infectiousness, the transmissibility and severity of the Beta variant were predicted in a mathematical modeling study [157]. The authors projected that the Beta VOC was 50% (95% CI, 20–113) more transmissible than previously circulating variants in South Africa, assuming perfect cross protection from previous exposures. This variant is 50% (95% CI: 20–113%) more transmissible than the Alpha variant in South Africa, according to a mathematical model [158]. In a study conducted by Yang and Shaman (2021), a model-inference system was developed to estimate epidemiological properties of new SARS-CoV-2 variants of concern using case and mortality data, accounting for under-ascertainment, disease seasonality, non-pharmaceutical interventions, and mass vaccination [101]. This study revealed that B.1.351 exhibits 32.4% (95% CI: 14.6–48.0%) increased transmissibility and 61.3% (95% CI: 42.6–85.8%) immune escape [101]. Brown et al. (2021) showed higher transmissibility of B.1.351 in Ontario, Canada (adjusted OR = 1.58: 95% CI: 0.93, 2.67) [107].

Early estimates suggested that the Beta variant had a transmission rate of 23–50% compared to ancestral lineages [159]. Seroprevalence estimates from blood donor surveys ranged from 32% to 62% across South Africa’s nine provinces, with a weighted national estimate of 47% following the second wave peak in January 2021 [160]. A report on SARS-CoV-2 variant transmission in Switzerland estimated a 50% increase in transmissibility [103]. In France, another study compared B.1.351 to B.1.1.7 to determine whether B.1.351 had a transmission advantage. In Île-de-France, the transmission advantage was 15.8% (95% CI: 15.5–16.2%), whereas in Hauts-de-France, the transmission advantage was 17.3% (95% CI: 15.9–18.7%) [161].

#### 2.3.2. Infectivity Rate

A study conducted by Motonzo et al. (2021) showed that at least two naturally occurring mutations in the SARS-CoV-2 RBM, L452R, and Y453F are able to bypass HLA-restricted cellular immunity and increase affinity for the viral receptor ACE2. The L452R mutation also improves the stability and viral infectivity of the S protein [162]. Reinfections are also attributed to this variant, which poses a public health risk. B.1.351 was discovered to be responsible for four reinfection cases in Luxembourg, Europe, according to Staub et al. (2021) [163]. In mouse and ferret models, the N501Y mutation has been linked to increased infectivity and virulence (~10^10^ copies/g 3 days after inoculation) [164].

According to a study conducted in Qatar, the B.1.351 variant was responsible for 413 (31.7%) of 1304 detected reinfections [165]. The median period between first infection and reinfection for reinfected people was 277 days. Reinfection increased the odds of severe illness by 0.12 times (95% CI: 0.03 to 0.31) compared to primary infection. The same study revealed that reinfections were 90% less likely to result in hospitalization or death than original infections [165].

#### 2.3.3. Disease Severity

A study co-ordinated by the ECDC in seven countries of the European Union (Cyprus, Estonia, Finland, Ireland, Italy, Luxembourg, and Portugal) revealed that the Beta variant was linked to a 3.6-fold increased risk of hospitalization and a 3.3-fold elevated risk of ICU admission [166]. In South Africa, this variant was linked to a 20% increase in in-hospital mortality in the second wave compared to the first wave [167].

In terms of hospitalization rate, severe illness rate, and fatality rate, the random effects of the Beta variant to wild-type virus are 2.16 (95% CI: 1.19–3.14), 2.23 (95% CI: 1.31–3.15), and 1.50 (95% CI: 1.26–1.74), respectively, according to a systematic review conducted by Lixin Lin in June 2021 [168].

A study conducted by Kleynhans et al. (2021) in South Africa showed that the age- and sex-adjusted seroprevalences of the rural and urban sites were 11.75% and 29.58%, respectively, during the first wave of infections (BD3), resulting in an ICR of 4.74% in rural and 1.93% in urban communities. Additionally, the same study revealed an IHR of 0.64% IHR in rural and 1.93% in urban communities; an in-hospital IFR of 0.12% and 0.16% in rural and urban communities, respectively; and an excess-death IFR of 0.43% and 0.12% in rural and urban communities, respectively [169]. Furthermore, in the same study, the second wave seroprevalence in rural and urban communities was determined to be 22.43% and 15.19%, respectively. The ICR was 3.71% and 3.67%, the IHR was 0.61% and 2.29%, the in-hospital IFR was 0.18% and 0.36%, and the excess deaths IFR was 0.65% and 0.50% for rural and urban communities, respectively [169].

In Qatar, researchers looked into the Beta (B.1.351) variant of COVID-19 disease. The Beta variant (B.1.1.7) had a 1.24-fold (95% CI: 1.11–1.39) higher risk of progressing to severe disease than Alpha (B.1.1.7). Beta had a 1.57-fold (95% CI: 1.03–2.43) increased risk of COVID-19 death [170].

#### 2.3.4. Affinity to Angiotensin-Converting Enzyme 2 (ACE2) Receptors

The RBD-hACE2 binding affinity of the Beta variant is 4.62 times higher than that of the SARS-CoV-2 RBD, according to the Ramanathan et al. (2021) [125]. The E484K and K417N mutations enhance binding affinity to human ACE2, whereas the combination of N501Y, E484K, and K417K improves binding even more [88].

#### 2.3.5. Viral Load

According to data from 871,604 PCR tests conducted by a large private clinical laboratory in France from 1 January to 24 March 2021, the viral load at symptom onset was higher in B.1.351 variants than in historical variants, with a Ct value of −1.15 (−1.57, −0.697) lower than preceding strains for B.1.351 [171]. According to a study by Radvak et al. (2021), K18-hACE2 animals infected with the B.1.351 variant had comparable or higher viral loads in multiple organs than those injected with a 10-fold higher dosage of NY (614G) or a 100-fold higher dose of WA [172]. A retrospective study by Golubchik et al. (2021) found that a sample of UK patients infected with the viral variant bearing the N501Y mutation had threefold higher inferred viral levels, demonstrating the high effectiveness of infection and transmission associated with the B.1.1.7 variant [173]. Infection of wild-type mice with SARS-CoV-2 B.1.351 variant indicated a possible novel cross-species transmission route [174]. Pan et al. (2021) discovered that a viral load was detectable in 5 of 12 trachea samples (10^3.425^, 10^2.286^, and 10^2.786^ copies/mL in hACE2 mice; 10^2^, and 10^2.562^ copies/mL in C57BL/6 mice) and 1 of 12 lung samples on day 7 (10^0.4114^ copies/mL in C57BL/6 mouse) [174].

#### 2.3.6. Reproduction Number (R_0_/R_t_)

The Beta variant’s R_t_ was 1.55 (95% confidence interval [CI]: 1.43–1.69), implying that it is 55% more transmissible than the wild-type strain [138]. Another study conducted in the United States revealed an effective reproduction number (R_t_) between 1 and 2 in most states [175]. In Bangladesh, the effective reproduction number for the B.1.351 variant was estimated to be 3.5 [176]. There is a scarcity of information concerning the Beta reproduction number.

#### 2.3.7. Vaccine Effectiveness and Vaccine Breakthrough

Multiple mutations in the receptor-binding region of the spike protein distinguish the B.1.351 variant [158], which is considered to be an escape variant from neutralizing antibody immunity generated during natural infection [177]. Although the Beta (B.1.351) variant was resistant to neutralization, it did not spread as much as the other variants [178]. Resistance hierarchy correlated to deletions in the N-terminal domain and alterations in the receptor-binding domain (RBD) of SARS-CoV-2, such as K417N/T, E484K, and N501Y [179].

Novavax, Johnson & Johnson, and AstraZeneca all conducted clinical studies that found a reduction in efficacy. Novavax clinical trials in the United Kingdom and South Africa revealed 89% efficacy and 60% efficacy, respectively [180,181]. The level of protection towards moderate to severe COVID-19 infection was 72% in the US, 66% in Latin America, and 57% in South Africa 28 days after immunization, according to Johnson & Johnson clinical trials. According to a study conducted by Chemaitelly et al. (2021) to examine the efficiency of Moderna’s vaccine against SARS-CoV-2 strains of concern in Qatar, its effectiveness against B.1.351 infection (Beta) was 61% after the first dose and 96% after the second dose [140]. According to Ho et al., the B.1.351 variant is markedly more resistant to neutralization by polyclonal antibodies produced in Pfizer (6.5-fold) or Moderna (8.6-fold) vaccine recipients [182].

When compared to ancestral D614G, data gathered from 48 healthcare workers (HCWs) in the United States 3–4 weeks after a second dose of either Moderna mRNA-1273 (n = 20) or Pfizer/BioNTech BNT162b2 (n = 28) indicate that Beta variants exhibited 1.2-fold (*p <* 0.001) reduced nAb titers [183]. The Ad26.COV2.S vaccination (J&J) produced cross-reactive binding and neutralizing antibodies against the B.1.351 strain (Beta) that were lower than those produced against the wild type but nonetheless offered robust protection against B.1.351 challenge, as determined by weight loss and pulmonary pathological score [184]. According to WHO reports, there was a reduction in neutralization by 5- to <10-fold for BNT162B2, mRNA-1273, and ChAdOx1 vaccines and by more than 10-fold for Ad26.COV2.S [149].

When compared to the original Wuhan strain, median pseudovirus neutralizing antibody titers generated by Janssen (Johnson & Johnson) vaccine: Ad26.COV2.S (Leiden, The Netherlands) were 5.0-fold lower against the B.1.351 variant [185].

Wang et al. (2021) [186] found that across all VOCs (excluding Omicron) and VOIs, Beta has the highest ability to break through vaccines, although its infectivity is relatively low (BFE change: 0.656 kcal/mol) in 20 COVID-19-devastated countries, including the United Kingdom, the United States, Denmark, Brazil, Germany, The Netherlands, Sweden, Italy, Canada, France, and India [186].

The Beta variant is the second VOC. Research indicates increased transmissibility of 23 to 50% and probable increased risk of severe disease and in-hospital mortality of 1.24 to 3.6-folds. High viral loads were detected, and the reproduction number was estimated to be around 1.55. For Pfizer, Moderna, and AstraZeneca vaccinations, neutralization was projected to be reduced by 5- to 10-fold, whereas Johnson & Johnson vaccine neutralization was reduced by more than 10-fold.

### 2.4. Gamma

Gamma (P.1) is another VOC, and it is defined as B.1.1.28.1. The P.1 variety is also known as 20J/501Y.V3 in the Nextstrain database and GR/501Y.V3 in the GISAID database. It was first discovered in November 2020 in Manaus, Brazil [100]. This variant is to blame for the high infection rate in of the second wave in Brazil [187]. The P.1 variant’s genome has accumulated a number of mutations (~17). The Gamma strain possesses some of the same spike protein mutations as the Alpha and Beta strains, allowing it to attach to human cells more easily [188]. This variant has about 12 mutations in the S glycoprotein [94]. The RBD region mutations are K417T, E484K, and N501Y. Aside from RBD, the S-glycoprotein mutations are T20N, R190S, D614G, P26S, D138Y, H655Y, L18F, and T1027I. This variant spread outside of Brazil and was found in other nations around the world. For example, it was found in numerous locations in Italy, according to Di Giallonardo et al. (2021), and was also discovered in Uruguay and Japan at the same time [189,190].

#### 2.4.1. Transmissibility

Three mutations in the receptor binding domain (RBD) of the Gamma VOC (K417T, E484K, and N501Y) can affect ACE2/RBD affinity, enhancing transmissibility or even antibody affinity [191,192]. Faria et al. estimated that the transmissibility of the Gamma variant in Brazil might be 1.4–2.2 times that of the wild-type virus, according to dynamic modeling that used genomic and mortality data, and that it evades 25–61% of the protective response resulting from infection with previously circulating variants [193]. The P.1 or Gamma variant of COVID-19 has been reported to be two times more transmissible than the original virus, and it has the potential to reinfect those who have already been infected with COVID-19 [194,195]. Given its links to a 70–240% increase in transmissibility, the P.1 lineage is considered a variant of concern [193].

Based on an epidemiological model-based fitting of data collected between 1 November 2020 and 31 January 2021, Coutinho 2021 discovered that the transmissibility of this variant is 2.5 times higher than that of the wild variant [196]. Based on another study using data collected between the beginning of November 2020 and the end of January 2021, Naveca predicted that P.1 might be at least twice as transmissible as the paternal lineage [197]. Gamma was estimated to be 1.7–2.4 times more transmissible in Canada than other local strains [188]. A study conducted by Hogan et al. found that the P.1 lineage was first identified in British Columbia around the end of February 2021, and it quickly grew to account for 39.4% of VOCs by March 2021 [198].

According to data collected in Italy between 18 February 2021 and 18 March 2021, this variant is up to 46% more transmissible than regional historical lineages, and the estimated relative transmissibility of P.1 ranged from 1.12 (95% CI: 1.03–1.23) in the case of complete immune evasion to 1.39 (95% CI: 1.26–1.56) in the case of complete cross protection [199]. P.1 exhibited a 43.3% (95% CI: 30.3–65.3%) increase in transmissibility and 52.5% immune escape (95% CI: 0–75.8%), according to a model-inference approach established by Yang and Shaman to evaluate epidemiological properties of new SARS-CoV-2 variants of concern [101]. A study conducted by Brown et el. (2021) in Ontario, Canada found high transmissibility of P.1 lineages (adjusted OR = 1.62, 95% CI: 1.21, 2.16) [137]. The Gamma variant has been detected in 74 countries throughout the world [200].

#### 2.4.2. Infectivity Rate

Globally, the prevalence of Gamma-infected cases has remained around 1% (total = 66,742) as of 13 January 2022; the locations with the highest reported prevalence were South and Central America (mainly Haiti, Brazil, Trinidad and Tobago, and French Guiana) [201]. The percentage of cases infected with Gamma among circulating lineages in the United States grew from 5.3% (3.7–7.6%) in the biweekly period of 11–24 April 2021 to 11.1% (8.7–13.9%) in the period of 23 May–5 June 2021 [202]. Another study conducted in Brazil revealed that the P.1 prevalence increased dramatically from 0% in November 2020 to 73% in January 2021, replacing prior lineages in less than 2 months [203]. In January 2021, a six-fold rise in the number of COVID-19 hospital admissions was reported in Brazil (compared to December) [187].

Studies have found a link between a lower Ct value (higher nucleic acid level) and a stronger ability to culture SARS-CoV-2, which is linked to enhanced infectivity [204,205]. Coutinho revealed that the average probability of reinfection is 6.4% based on data gathered between 1 November 2020 and 31 January 2021 [196]. Researchers found that reinfections in the Amazonian region and the Brazilian state of São Paulo were caused by the Gamma strain [206,207].

#### 2.4.3. Disease Severity

The Gamma variant was linked to a 2.6-fold higher risk of hospitalization and a 2.2-fold higher risk of ICU admission, according to a study coordinated by the ECDC in seven European Union countries (Cyprus, Estonia, Finland, Ireland, Italy, Luxembourg, and Portugal) [166].

Between epidemiological week 38 of 2020 and epidemiological week 10 of 2021, Funk et al. compared Gamma (n = 325) to non-VOC instances (n = 3348) recorded in seven European nations (Cyprus, Estonia, Finland, Ireland, Italy, Luxembourg, and Portugal). Infection with Gamma was found to be associated with significantly higher odds of hospitalization (AOR: 2.6, 95% CI: 1.4–4.8) and ICU admission (AOR: 2.2, 95% CI: 1.7–2.8) but not mortality when compared to non-VOCs [208].

In Rio Grande do Sul, Brazil, an increase in the risk ratio (RR) of severe sickness and mortality was reported in the second wave (epidemiological weeks 5 to 8 of 2021; n = 80,912). For all patients, the RR of severe illness in the second wave compared to the first was 1.7 (95% CI: 1.64–1.76); *p* < 0.0001. The RR of mortality was 2.06 (95% CI: 1.94–2.19); *p* < 0.0001 [209].

#### 2.4.4. Affinity to Angiotensin-Converting Enzyme 2 (ACE2) Receptors

The N501Y, K417N, and E484K mutations, which are likewise present in the Alpha and Beta forms, have been linked to higher binding affinity to human ACE2 and transmissibility [104,187,210]. P.1, like other VOCs, is linked to enhanced binding to the human ACE2 receptor, according to Faria et al. (2021) [193].

#### 2.4.5. Viral Load

In a peer-reviewed modeling study in Brazil, the viral load was shown to be roughly ten times (Ct = 19.8 vs. 23.0; *p* < 0.0001) higher than in non-Gamma patients [195]. When comparing June 2020 to February 2021, there was a substantial decrease in the median Ct values (25 to 22) from nasopharyngeal swab samples examined by RT-qPCR (*p* < 0.0001) in Brazil [211].

To et al. (2022) demonstrated that immunization with both 5 and 25 g doses of S significantly lowered Gamma viral load in both the upper and lower respiratory tracts, signifying a reduced likelihood of viral transmission, even in macaques with minor symptoms [212]. There has been little research on the viral load of the Gamma variant.

#### 2.4.6. Reproduction Number (R_0_/R_t_)

Using phylogenetic approaches, Naveca and colleagues (4) determined that the P.1 variant has a 2.2 times higher effective reproduction number (R_t_) than the parental lineage [203]. The Gamma variant has an effective reproduction number (R_t_) of 2.6, according to a peer-reviewed modeling study conducted in Brazil based on data collected prior to 13 January 2021 [197]. The R_t_ of Gamma was estimated to be 38% (95% CI: 3–17%) greater than that of non-VOCs, 10% higher than that of Alpha, and 17% higher than that of Beta in another peer-reviewed genomic epidemiology study [159]. Oróstica et al. (2021) discovered that the reproduction numbers of Gamma variant were approximately 16% higher than those of the Alpha variant [213]. There is a scarcity of information about the Gamma reproduction number.

#### 2.4.7. Vaccine Effectiveness and Vaccine Breakthrough

The AstraZeneca vaccination provides good protection against the Gamma variant. During a period when the Gamma variant was prevalent, the two-dose regimen was 77.9% effective against COVID-19 by (95% CI: 69.2–84.2), 87.6% effective against hospitalization (95% CI: 78.2–92.9), and 93.6% effective against death (95% CI: 81.9–97.7) in Brazil [214]. The Gamma lineage has been discovered to have a high level of resistance to neutralizing antibodies due to its mutations [215]. The P.1 variant was much less resistant to naturally acquired or vaccine-induced antibody responses than the B.1.351 variant, indicating that alterations beyond the receptor-binding domain have an impact on neutralization [191]. On 14 January 2021, scientists in Brazil reported that a coronavirus vaccine developed by Sinovac (CoronaVac) was 50.38% effective when tested in 12508 volunteers, all of whom were health professionals who had direct contact with the coronavirus [216]. According to Shapiro et al., the average vaccine effectiveness for the VOC γ (P1) is 61% (95% CI: 43–73%) [217]. According to the WHO, there was a reduction in neutralization by 2 to <5-fold for BNT162B2, mRNA-1273, ChAdOx1, and Ad26.COV2.S vaccines [149].

Lineage of SARS-CoV-2 antibodies produced in response to polyclonal stimulation against previously circulating SARS-CoV-2 variants may not be able to neutralize P.1 (Gamma). The neutralizing ability of plasma from people who had previously been infected with SARS-CoV-2 was 8.6 times lower against P.1 isolates [218]. When compared to the original Wuhan strain, the median pseudovirus-neutralizing antibody titers generated by Ad26.COV2.S (Johnson & Johnson) were 3.3-fold lower against the P.1 variant [185].

In summary, the third variant of concern is Gamma. In comparison to the wild type, higher transmissibility and infectivity were noted. Hospitalization risk, disease severity, and viral load levels have all been reported to be higher. The effective reproduction number for the Gamma variant was reported to be 38% higher than that of non-VOCs, 10% higher than that of Alpha, and 17% higher than that of Beta, according to global data. Pfizer, Moderna, AstraZeneca, and Johnson & Johnson vaccines were found to have a 2- to 5-fold reduction in neutralization.

### 2.5. Delta

The Delta variant (B.1.617.2) was detected for the first time in India in October 2020, and by December 2021, it had become the widely dominant variant, remaining so until January 2022 [219]. In comparison to the first reported COVID-19 strain (Alpha strain), the Delta variant includes 23 mutations [220]. The spike protein has twelve mutations. RBD mutations in B.1.617.2 are L452R and T478K. Other than the RBD, significant mutations in the S glycoprotein include T19R, G142D, D614G, P681R, 157del, R158G, 156del, and D950N [94]. This variant’s mutation at position P681 leads to SARS-CoV-2 transmission and infection [221,222]. During the second half of 2021, the Delta variant (B.1.617.2) became the most common variant in much of the world [223] and has been detected in more than 100 countries.

Additionally, the ‘Delta Plus’ strain is a mutated version of the more virulent B.1.617.2, or ‘Delta’, strain that caused the second wave of infections in India [224]. The K417N mutation in the spike protein of the SARS-CoV-2 virus that produces the COVID-19 disease has been identified as B.1.617.2.1. There were 205 Delta Plus (AY.1) lineage sequences found worldwide as of June 18, 2021, with the United States and the United Kingdom accounting for more than half of the cases [224]. The Delta Plus strain (AY.1) demonstrated “increased transmissibility, higher binding to lung cell receptors, and probably diminished monoclonal antibody response”, according to the Indian SARS-CoV-2 Genomics Consortium [159,224,225,226].

#### 2.5.1. Transmissibility

Delta appears to be more transmissible than Alpha and other SARS-CoV-2 variants due to its rapid spread in various parts of the world [227]. Because of differences in underlying immunity, control measures, behaviors, and demography, estimations of Delta’s transmissibility may vary amongst communities. A variant that is more prone to produce vaccine breakthroughs, for example, may have a larger observed transmissibility advantage in communities with higher vaccination rates because it can spread to more people [227]. Furthermore, estimations might be influenced by the quality and volume of data collected by sequencing programs in various places.

Vaccine escape and higher transmissibility have been linked to the L452R and T478K mutations found in B.1.617.2 [228]. The Delta variant is twice as transmissible as the Alpha variant, according to the CDC [98]. In comparison to the Alpha variant, which was twice as contagious as the original Wuhan strain, the American Society for Microbiology reported a 40–60% rise in transmissibility of the Delta variant [229]. India was the first country to record more than 400,000 new COVID-19 cases in the first 24 h on 1 May 2021 [230]. According to statistics from Public Health England’s weekly report on COVID-19 variant cases, the number of instances of the Delta variant in the UK increased by 33,630 from 11 June to 18 June 2021, for a total of 75,953, representing a 79% increase [231]. Delta is predicted to be 40–80% more transmissible than Alpha, which was itself more transmissible than the SARS-CoV-2 lineages previously in circulation, according to studies conducted in the United Kingdom [232]. In many other countries, notably the United Kingdom and the United States, B.1.617.2 rapidly became the most prevalent variant. B.1.617.2 first appeared in the UK in mid-April, accounting for 95% of all new cases [233], 99% of new cases in the US [234], and 70% of cases in Lisbon [233]. Earnest et al. (2022) assessed variant-specific effective reproduction numbers and estimated that Delta is 63–167% more transmissible than Alpha (range across the United States) based on data collected between April and mid-July 2021 [227]. Based on statistics from India and the United Kingdom, the World Health Organization estimated a 55% rise in Delta transmissibility [159].

A study conducted by Mahase et al. (2021) found that B.1.617.2 is 60% more transmissible than B.1.1.7 [235]. After adjusting for the vaccination status of the index cases, as well as sex, ethnicity, IMD, age group, and the number of household contacts, there was a 70% increase in the likelihood of household transmission associated with infection with the SARS-CoV-2 Delta variant compared to Alpha [236]. Another modeling study from the United Kingdom found that S-gene positives, which are a surrogate for the Delta VOC, may have 1.4-fold increased transmissibility compared to other lineages [237].

It has been suggested that the incubation period has shortened since the emergence of the more transmissible Delta variant. According to Kang et al. (2021) the average incubation period is 5.8 days (95% CI: 5.2–6.4), with 95% of infected people displaying symptoms by 11.5 days [238]. Zhang et al. also determined a mean incubation period of 4.4 days (95% CI: 3.5–5.0) in another study, which appears to be slightly shorter [239].

#### 2.5.2. Infectivity Rate

Several studies have shown that the L452R enhances infectivity by stabilizing the association between the SARS-CoV-2 spike glycoprotein and the human ACE2 receptor [240,241]. The Delta variant of the SARS-CoV-2 strain is 60% more infectious than the original Wuhan strain [242].

During a three-month period (October to December 2020), the Delta variant infected more than 26% of the Indian population and the immune-evasive properties of the Delta strain are most likely to blame for the high transmission rate [243]. Additionally, Delta cases increased in the United States from 0.6% (0.4–0.8%) during the period from 11 to 24 April 2021 to 11.1% (7.6–13.8%) during the period of May 23–June 5 (2021), then surging to 83.2% (79.2–86.8%) during the period of 4–17 July 2021 [202].

Data collected retrospectively from the Guangdong, China outbreak in May–June 2021 by Kang et al. (2021) revealed that the secondary attack rate among close contacts of Delta cases was 1.4%, and 73.9% (95% Cl: 67.2%, 81.3%) of transmissions occurred before onset. Index cases who did not receive vaccination (OR: 2.84, 95% Cl: 1.19, 8.45) or who only received one dose of vaccination (OR: 6.02, 95% CI: 2.45, 18.16) were more likely to infect their contacts than those who had received two doses of immunization [9]. In China, the generation time between primary and secondary cases was 2.9 days, which is shorter than that of previous variants [239].

#### 2.5.3. Disease Severity

Clinical research has shown that the emergence of the Delta variant can cause damage to the lungs, liver, gastrointestinal system, vascular system, neurological system, heart, pancreas, kidneys, and other organs of infected people, resulting in a variety of clinical manifestations, such as cough, pneumonia, dyspnea, ARDS, respiratory failure, liver damage, abdominal pain, disseminated intravascular coagulation, myocardial damage, and pancreatic toxicity [57,244,245]. In terms of disease severity, a Singapore study indicated that infection with the Delta variant was linked to a greater likelihood of oxygen treatment, ICU hospitalization, and death after adjusting for age and gender (adjusted odds ratio: 4.90 (95% CI: 1.43–30.78)) [246]. The Delta variant appeared to cause roughly 60% more hospitalizations than previous variants, according to Mahase et al. (2021), and this was concerning [235]. In England, the Delta variant had a case fatality rate (CFR) of 0.2% [247].

Between 29 March and 23 May 2021, Katherine Twohig and colleagues found an elevated hospitalization risk for Delta variant infections (hazard ratio: 2.26 (95% CI: 1.32–3.89)) compared to Alpha variant infections in England [248]. Similar results were obtained in a study conducted in Denmark by Bager (2021), who found the Delta variant was associated with an increased risk of hospitalization (risk ratio 2·83 (95% CI: 2.02–3.98] [115]. Additionally, according to a study conducted in Scotland, the Delta variant increased the chance of hospitalization by double when compared to the Alpha variant. The same study also showed that the Delta form was more common in the younger and the most socioeconomically affluent quintiles of the population [249]. It is of note that this variant expanded rapidly in UK primary and secondary schools [250]. In comparison to the original virus, the Delta variant has a 108% higher risk of hospitalization, a 235% higher risk of ICU admission, and a 133% higher risk of mortality [243].

In Canada, a retrospective cohort study [251] found that patients infected with the Delta variant had an adjusted risk of hospitalization of 120% (93–153%), a risk of ICU admission of 287% (198–399%), and a risk of death of 137% (50–230%), all of which were significantly higher than those of other VOCs. Furthermore, with the Delta variant, the odds ratios (OR) for hospitalization, ICU admission, and mortality were 2.20 (95% CI: 1.93–2.53), 3.87 (95% CI: 1.5–3.3), and 2.37 (95% CI: 1.50–3.30), respectively [251].

#### 2.5.4. Affinity to Angiotensin-Converting Enzyme 2 (ACE2) Receptors

A recent study conducted by Khatri et al. (2022) showed that the Delta synthetic mutant exhibited a considerable increase in pseudovirus entry, fusion, and infectivity [252]. A leucine-to-arginine substitution at position 452, which is present in Delta but not in the Omicron variant, is known to boost affinity for ACE2 receptors located on the surface of a variety of human cells, including lung cells [6]. Delta spike was found to fuse membranes more effectively at low levels of the cellular receptor ACE2; its pseudotyped viruses also attack target cells much faster than the other five variants (G614, Alpha, Beta, Gamma, and Kappa), which could explain its increased transmissibility [253]. Compared to D614G alone, Deng X et al. found that pseudoviruses containing the L452R mutation had a 6.7–22.5-fold higher entry efficiency into host cells in 293T cells and 5.8–14.7-fold higher entry efficiency in human airway lung organoids (HAOs) (293T cells and HAOs can stably express ACE2) [254]. L452R and T478K, which are among 17 mutations in the Delta spike protein, are responsible for the virus’s improved capacity to avoid a host’s immune response [255]. Two mutations in the receptor-binding domain of B.1.617 (Delta), which interacts with the ACE2 receptor and is the primary target of neutralizing antibodies, are present in the spike protein of B.1.617 (Delta). The B.1.617 spike protein can marginally increase the efficiency of entry into lung cells, indicating that soluble ACE2 and camostat can both impede entry [256].

#### 2.5.5. Viral Load

Research data suggest that the L452R mutant might circumvent HLA-24-mediated cellular immunity, increase viral infectivity, and possibly increase viral replication [257]. The Delta variant had much larger viral loads, i.e., a lower PCR cycle threshold (Ct) value, and a longer Ct ≤ 30 [246]. According to a survey of 292,284 people tested in France from 14 June to 30 July 2021, infection with the Delta variant was also related to lower C_t_ levels (effect size: 6.7; 95% CI: 7.1 to 6.3) when compared to non-Delta (mainly Alpha) variants [258]. During their respective emergences, Delta infections produced 6.2 (95% confidence interval 3.1–10.9) times more viral RNA copies per milliliter than Alpha infections [227].

According to Teyssou et al. (2021), the Delta SARS-CoV-2 variant has a higher viral load than the Beta and historical variants, which could indicate greater infectivity [259]. In the case of the ORF1ab target gene, the Delta variant had a VL ten times higher than that of historical variants (median 7.83 log_10_ copies/mL [6.3–8.83]) (*p* < 0.0001). The Delta variant had a twofold difference in ORF1ab VL compared to the Alpha and Beta variants (median 7.58 (5.79–8.69) and 7.42 (6.06–8.38), respectively). The Delta variant (median 7.69 (6.58–8.94)) had a considerably higher VL (5-fold) than the historical variants (median 7.02 (5.26–8.15)) for the N gene (*p* < 0.0001). There was also a significant 2.5-fold difference in VL levels between the Delta and Beta variants (median 7.26 (6.10–8.37); *p* < 0.05) [259].

Data gathered retrospectively from an outbreak in Guangdong, China, in May–June 2021 by Kang et al. revealed that the mean estimates of the latent period and incubation period were 4.0 days and 5.8 days, respectively. Delta infections showed a much higher viral load than wild-type infections, with a median Ct value for the Delta variant’s N gene of 23.0 (IQR: 19.3–28.6), which is significantly lower than that of wild-type N genes (median: 36.5, IQR: 33.0–40.0) [9].

Another study conducted by Li et al. in China showed that relative viral loads of cases infected with the Delta variant (n = 62, Ct = 24.00 for the ORF1ab gene, IQR 19.00–29.00) were 1260 times higher than those of cases infected with clade A/B viruses (n = 63, Ct = 34.31 for ORF1ab gene, IQR 31.00–36.00) [260]. Furthermore, results suggest that the Delta variant may be more infectious early in course of the illness [260].

Individual serial Ct values were used as a surrogate marker for viral load. Unvaccinated and fully vaccinated patients had the same first median Ct value (unvaccinated median Ct: 18.8 (14.9–22.7), vaccinated” 19.2 (15.2–22.2)). However, compared to unvaccinated individuals, fully vaccinated patients had a faster rate of increase in Ct value over time, implying a more rapid decline in viral load (coefficient estimates for interaction terms ranged from 6.62 (standard error 3.364) to 9.30 (standard error 3.04) for each of the interaction terms [261].

Delta had substantially higher infectivity than Alpha based on the number of infectious units per quantity of viral E-gene RNA (5.9-fold increase; *p* < 0.0001) or subgenomic E RNA (14.3-fold increase; *p* < 0.0001) in clinical specimens identified as SARS-CoV-2-positive at the University of Washington Virology Laboratory between March and August 2021 in the United States [262].

Between 28 November 2020, and 11 August 2021, data from a longitudinal set of 19,941 SARS-CoV-2 viral samples collected from 173 participants as part of the National Basketball Association’s occupational health program revealed no significant differences in mean peak viral load (with a lower peak cycle threshold (Ct) indicating a higher viral load), proliferation duration, clearance duration, or duration of acute infection of either the Alpha or Delta variant [263]. In the same study, there no noticeable difference between vaccinated and unvaccinated patients was observed in terms of mean peak viral load or proliferation length [263].

Vaccinated patients had 2.8 times fewer RNA genome copies than unvaccinated patients (0.44 log_10_, *p* = 0.0002), according to a study conducted by Puhach et al. (2022) based on nasopharyngeal swabs obtained from symptomatic individuals in the outpatient testing center of the Geneva University Hospital. In vaccinated patients, the reduction in infectious VL was more significant (4.78 fold, 0.68 log_10_, *p* < 0.0001) [264]. Additionally, infectious VLs were significantly higher in unvaccinated patients compared to vaccine breakthrough cases (8.12 times, 0.91 log_10_, *p* < 0.0001), indicating that vaccinated patients have significantly lower infectious VLs [264].

#### 2.5.6. Reproduction Number (R_0_/R_t_)

A study conducted by Zhang et al. in China between May and June 2021 revealed a basic reproduction number R_0_ = 3.2 (95% CI: 2.0–4.8) [239]. In another study conducted in China after May 2021, R_0_ was estimated to be between 4.04 and 5.0 [265]. Furthermore, R_0_ was assessed to be equal to 5.08, whereas the R_0_ of the ancestral strain was estimated to 2.79 in a study conducted by Liu et al. in China in July 2021 [266]. R_0_ estimations in England based on SPI-M-O modeling the were between 5 and 8 before the June 2021 election (SPI-M-O). Mackie 2021 discovered similar results in a UK investigation, wherein R_0_ was determined to be equivalent to 5.2 [267]. According to a study conducted by Burki et al. in the United Kingdom in July 2021, the R_0_ was determined to be between 6 and 7, implying that each infected person in the United Kingdom can spread the virus to 6 to 7 individuals [268]. Earnest et al. (2022) estimated the mean effective reproduction number (R_t_) for Delta to be 1.40, ranging from 1.27 in New Hampshire to 1.65 in Vermont in the United States, based on data collected between April and July 2021 [227].

#### 2.5.7. Vaccine Effectiveness and Vaccine Breakthrough

The efficacy of vaccines varied depending on the vaccine type. A study conducted by Bernal et al. revealed that a single dose of BNT162b2 (BioNTech (Manhattan, NY, USA), Pfizer vaccine) or ChAdOx1-S (AstraZeneca, Cambridge, UK) vaccines had similar efficacies against the Delta and Alpha variants, with 30.7% and 48.7%, respectively [269]. The same study also reported that the BNT162b2 vaccine is 88% effective against Delta infection after two doses. Two AstraZeneca doses, on the other hand, were shown to be less effective (67%) against Delta variant infection [269]. Tang et al. (2021) found that both BNT162b2 and mRNA-1273 vaccines were effective in avoiding Delta hospitalization and death in Qatar’s population, despite reduced effectiveness in preventing infection, notably for the BNT162b2 immunization [270]. The same study revealed that the effectiveness of BNT162b2 against any symptomatic or asymptomatic Delta infection was 45.3% (95% CI: 22.0–61.6%) ≥ 14 days after the first vaccine dose and only 51.9% (95% CI: 47.0–56.4%) ≥ 14 days after the second dose, with 50% of fully vaccinated individuals receiving their second dose before 11 May 2021. mRNA-1273 effectiveness was 73.7% (95% CI: 58.1–83.5%) and 73.1% (95% CI: 67.5–77.8%) ≥ 14 days after the first and second doses, respectively [270]. Based on data gathered between 27 July 2020 and 14 November 2020, Polack et al. (2020) found that a two-dose regimen of BNT162b2 (30 μg per dose administered 21 days apart) provided 95% (95% CI: 90.3 to 97.6) protection against the COVID-19 Delta variant in people aged 16 years and older [271]. After the first dose of the Astra Zeneca ChAdOx1-S vaccination, the Delta attack rate was 35.7% (5/14) for staff and 81.3% (13/16) for residents, showing that persons who had received only one dose within three months offered less protection. However, hospitalization was infrequent, and no deaths were reported, suggesting that a single dose of vaccination could protect against severe disease after infection with the Delta form [272].

The mRNA-1273 vaccination was found to be 94.1% (95% CI: 89.3 to 96.8%; *p* < 0.001) effective in avoiding COVID-19 sickness due to the Delta variant, including severe disease, in research conducted in the United States [273].

Between 7 September and 24 November 2020, data from 21977 adults who were randomly assigned to the vaccine group (n = 16 501) ((rAd)-based vaccine, Gam-COVID-Vac (Sputnik V)) or the placebo group (n = 5476) for a randomized, double-blind, placebo-controlled, phase 3 trial at 25 hospitals and polyclinics in Moscow, Russia, revealed a vaccine efficacy of 91.6% (95% CI 85.6–95.2) [274].

In vitro research and molecular epidemiology were combined in a study conducted by Mlcochova et al. (2021) which confirmed improved replication fitness and decreased sensitivity of SARS-CoV-2 B.1.617.2 (Delta) to neutralizing antibodies [275]. Delta eludes neutralizing mAbs and polyclonal antibodies generated by earlier SARS-CoV-2 infection or immunization in a partial but considerable manner. Sera from individuals who had received one dose of either the Pfizer or AstraZeneca vaccination revealed a slight effect on the Delta variant. In 95% of people, two doses produced a neutralizing response, with titers against Delta being three to five times lower than those against Alpha [276]. Wang found that B.1.351 was more resistant to neutralization of convalescent serum (by a factor of 2) and vaccine serum (by a factor of 2.5–3.33) than the wild-type virus in his investigation [277]. A structure–function investigation of B.1.351 was performed by Zhou and colleagues from Oxford, UK, utilizing a large cohort of convalescent and vaccine serum samples. Receptor-binding domain alterations result in stronger ACE2 binding and widespread resistance to monoclonal antibody neutralization. E484K is the most common mutation, whereas K417N and N501Y work synergistically against some major antibody classes. Both the Pfizer and AstraZeneca vaccines reduced the neutralizing titers for B.1.351 by eight to ninefold [178]. Furthermore, based on the WHO reports, there was a reduction in neutralization by 5 to <10-fold for BNT162B2, by 2 to <5-fold for mRNA-1273 and ChAdOx1 vaccines, and by less than 2-fold for Ad26.COV2.S [149].

Increased rates of breakthrough infections not only have a detrimental influence on global public health security but also heighten speculation about vaccine failure, increasing societal concern [278,279]. From January to July 2021, clinical chart reviews were conducted for all Delta- (107) and Alpha (1482)-infected patients diagnosed at the Johns Hopkins laboratory, and the Delta variant was found to cause a significant increase in confirmed breakthrough infections, particularly in comparison to the Alpha variant (28% vs. 12.4%, respectively) [280].

Although the secondary attack rate among fully vaccinated household connections was high, at 25%, it was lower than the rate among unvaccinated contacts (38%). Individuals who had been fully vaccinated when they developed breakthrough infections, on the other hand, had viral loads equivalent to those who had not been vaccinated [281].

In conclusion, the Delta variant of the novel coronavirus, the fourth VOC, is a highly transmissible variant of the virus that is perhaps 60–70% more transmissible than the Alpha variant and 60% more infectious than the wild-type virus. When compared to all other variants, the increased risk of hospitalization, admission to an intensive care unit, and possible increased risk of fatality was the highest. A high viral load was also reported. The incubation period has shortened since the emergence of the more transmissible Delta variant. Each sick person has the potential to infect as many as six people. Although this variant is more resistant to neutralization than the Alpha, Beta, and Gamma variants, there is evidence that it continues to protect against markers of severe disease, such as hospitalization and fatality.

### 2.6. Omicron

The World Health Organization (WHO) classified Omicron (B.1.1.529) as the fifth SARS-CoV-2 VOC on 26 November 2021 [282,283]. Omicron is the world’s fastest-spreading virus in humanity [284]. Only a month after being discovered in southern Africa, the new coronavirus variant was already causing havoc in several countries around the globe, with more cases than ever before [285].

The spike (S) protein in Omicron has an unusually high number of mutations. The first genome-sequencing data from Botswana revealed 50 mutations in this novel variant, of which, 32 mutations are in the S protein [286]. In the RBD of the S protein of Omicron variant, there are many amino acid alterations, including K417N, N440K, G446S, S477N, T478K, E484A, Q493K, G496S, Q498R, N501Y, and Y505H. The most mutated variant SARS-CoV-2, is now circulating in more than 150 countries/territories [287]. The high mutation rate in the S protein is a major concern, with the knowledge that many of the mutations have been reported to enhance the risk of infection while avoiding immune response.

Of more concern, the Omicron variant has evolved into five lineages: BA.1, BA.2, BA.3, BA.4, and BA.5 [288,289]. As of January 2022, Omicron’s new subvariant, BA2, was causing concern because this new variant spreads more quickly and infects the lungs more efficiently. The BA.2 lineage has been labelled “Stealth Omicron” because it differs from the “standard” variant in that it lacks the characteristic SGTF-causing deletion (H69del) [288].

According to a new report from the World Health Organization on Omicron sub-variant BA.2, the strain accounted for 21.5% of all new Omicron cases examined globally in the first week of February 2022 [290]. Bangladesh, Brunei, China, Denmark, Guam, India, Montenegro, Nepal, Pakistan, and the Philippines are the ten nations where BA.2 is now widespread. The fact that BA.2 has been steadily increasing in prevalence across various countries shows that it has a growth advantage over other circulating variants (BA.3) [291]. One explanation for the rise of BA.2 is that it is even stronger than BA.1 at overcoming immunity, which could include the protection provided by a BA.1 infection.

According to data obtained by the WHO on 14 February 2022, the number of new COVID-19 cases reported worldwide has continued to diminish, with 2.7 million new cases recorded in the week of 7 February, a 16% decrease from the previous week [287]. Additionally, the number of new deaths has also decreased, with 81,000 new deaths reported in the week of 7 February, down 10% from the prior week [290]. The UK Health Security Agency classified the variants BA.4 and BA.5 as Omicron lineage variants [292]. The first incidence of the BA.4 variant was reported in South Africa in January 2022. BA.4 and BA.5 have similar mutations and share the same F486V amino acid mutation; however, the BA.5 variant has additional characteristics [293].

The Omicron variant is hypothesized to have evolved in a single immunocompromised human patient or a persistently infected COVID-19 patient over weeks or even months with little surveillance, or it may have developed in a non-human animal and recently spilled back into humans due to its rapid mutation rate [294].

#### 2.6.1. Transmissibility

Multiple pieces of evidence suggest that Omicron is extremely transmissible, spreading much quicker than any other SARS-CoV-2 variant [295,296]. High transmissibility has been linked to the N679K, N501Y, P681H, N679K, and D614G mutations [220,297]. According to a New York Times database, health officials around the world reported an average of 2.1 million cases during the first week of January 2022, which is more than three times the rate of the preceding two weeks [298]. According to the WHO, more than 22 million new cases and 59 000 deaths were confirmed in the last week of January 2022 [296]. Zeng et al. (2021) reported that Omicron has a significantly enhanced efficiency of cell-to-cell transmission, with 4.8-fold higher levels than D614G and other variants [183]. On 8 November 2021, this variant was discovered in South Africa, and within two weeks, more than 70% of sequenced samples from Gauteng province in South Africa were Omicron [299]. By 11 December 2021, the United States had reported that the Omicron variant was responsible for nearly 7% of COVID cases. In the next two weeks, this rate quickly increased to around 23% and 60%, respectively [300]. The variant has now been identified in all 50 states and became the dominant variant within just three weeks since its discovery in the United States [299]. On 4 January 2022, the United States alone recorded more than one million cases [301]. It was also reported that every 2–3 days, Omicron variant cases doubled in the United Kingdom [302]. A study conducted by Japanese scientists showed that the Omicron variant of COVID-19 is 4.2 times more transmissible in its early stages than the Delta variant [303].

Qatar has been facing a large Omicron wave that began in December 2021 and peaked in mid-January 2022 [304]. Around January 2022, more than 80% of cases were assigned to BA.2 variant [305].

Early data from South Africa revealed that the Omicron variant was 3.3 times (95% CI: 2.0, 7.8) more transmissible than the Delta variant based on the relative exponential growth rate from 18 October to 30 November 2021 [306]. Lyngse et al. (2021) found that the Omicron variant is 2.7 to 3.7 times more transmissible than the Delta variant in a Danish household study [307]. The United Kingdom and the United States had recorded the most SARS-CoV-2 Omicron variant infections as of 3 February 2022, with more than 345 and 282 thousand cases, respectively [308]. Furthermore, a Danish study revealed that BA.2 households had a higher risk of transmission from unvaccinated primary cases than BA.1 households, with an OR of 2.62. (95% CI: 1.96–3.52) [309].

According to a recent study by Lyngse et al. (2022) of Omicron spread in more than 8000 Danish families, the growth of BA.2 is due to a combination of reasons [309]. Unvaccinated, fully vaccinated, and booster-vaccinated people were all more susceptible to BA.2 than BA.1, showing that BA.2 has a higher transmissibility than BA.1 [309].

Bi et al. estimated the likelihood of undetected Omicron transmission in LMICs by early December based on simulations [310]. Turkey (99.99%), Pakistan (99.95%), and Serbia (99.81%) had the highest estimated risk among the LMICs assessed, followed by Nepal (87.98%), Bangladesh (84.86%), and the Dominican Republic (82.21%). According to the same study, if these countries implement non-pharmaceutical treatments that restrict transmission by 80%, the likelihood of undetected emergence drops from 12.02 percent to 80.77 percent among the 25 LMICs [310].

#### 2.6.2. Infectivity Rate

The Omicron variant’s high number of mutations in the S protein could help the virus elude infection-blocking antibodies and other immune responses, such as the T-cell response [286]. The lysine-to-asparagine alteration at position 417 (not found in all Delta sequences but common in the Beta variant) has been linked to S-protein structural changes that may aid in immunological escape. The threonine-to-lysine substitution at position 478 is likely to boost the electrostatic potential and steric interference of the residue, perhaps increasing RBD binding affinity and allowing immunological escape [286]. In addition to the other mutations, the Omicron variant has a three amino-acid deletions in ORF1a at L3674-, S3675-, and G3676-, which may aid in innate immune evasion by impairing the ability of infected cells to destroy viral components [311,312]. Additionally, several studies have shown that the binding free energy (BFE) between the S RBD and the ACE2 is proportional to the infectivity of the virus [313,314]. According to the Centers for Disease Control and Prevention (CDC), the Omicron variant accounted for 73% of new infections in the United States as of December 20, 2021—up from 13% the week before [98]. According to Gozzi et al. (2022) Omicron gained dominance (i.e., more than 50% of infections) in mid-November 2021 and supplanted Delta by early December [315].

A retrospective analysis of presumed reinfections among South Africans revealed a temporally juxtaposed Omicron emergence with a higher risk of reinfection, which was not seen in previous waves associated with Beta or Delta variants, implying that the Omicron variant is linked to a unique ability to overcome natural immunity from prior infection [316]. People who had previously been infected with COVID-19 were more likely to get infected with Omicron than with previous variants, with 9.5% of those affected having previously been infected in the UK [317]. Using an artificial intelligence (AI) analysis, Chen et al. (2021) reported that Omicron is potentially ten times more contagious than the original virus and nearly twice as infectious as the Delta variant [318]. Additionally, the Imperial College London COVID-19 response team found that Omicron was associated with a 5.41 (95% CI: 4.87–6.00)-fold higher risk of reinfection than Delta after controlling for vaccine status, age, sex, ethnicity, asymptomatic status, region, and specimen date [319]. According to Callaway and Ledford, over the same time period, Omicron is estimated to have infected three to six times as many people as Delta [320]. Furthermore, a study based on national data in Scotland indicated that Omicron has a ten times higher incidence of potential reinfection than Delta [321]. Lyngse et al. (2021) observed a 1.17 (95% CI: 0.99–1.38) times higher secondary attack rate (SAR) for unvaccinated individuals, a 2.61 (95% CI: 2.34–2.90) times higher SAR for fully vaccinated individuals, and a 3.66 (95% CI: 2.65–5.05) times higher SAR for booster-vaccinated individuals in a study comparing households infected with the Omicron to those infected with the Delta VOC [307].

Moreover, according to a study conducted in Denmark, the secondary attack rate (SAR) in families infected with Omicron BA.1 and BA.2 was estimated to be 29% and 39%, respectively [309]. When compared to BA.1, BA.2 was found to be associated with increased susceptibility to infection in unvaccinated individuals (OR 2.19; 95% CI: 1.58–3.04), fully vaccinated individuals (OR 2.45; 95% CI: 1.77–3.40), and booster-vaccinated individuals (OR 2.99; 95% CI: 2.11–4.24) [309].

#### 2.6.3. Disease Severity

When compared to the Delta variant, the Omicron variant replicates more than 70 times more efficiently in the human bronchus but less efficiently in lung tissue [322], implying reduced disease severity [282]. In comparison to Delta and other strains, Omicron appears to prefer infecting and multiplying in the upper respiratory tract [296]. According to the UK Health Security Agency, people infected with the Omicron variant of SARS-CoV-2 are between 31 and 45% less likely to seek emergency care and 50–70% less likely to be admitted to hospital than those infected with the Delta variant. Between 14 November and 4 December, 6.3% of intensive care beds in South African hospitals were occupied, according to the World Health Organization, which was “very low” when compared to the same period in July, when the country was experiencing a peak associated with the Delta variant. Additionally, a matched cohort study on gender, age, vaccination status, health region, and onset date found that Omicron cases had a 65% lower risk of hospitalization or death (hazard ratio, HR = 0.35, 95% CI: 0.26, 0.46) than Delta cases, whereas the risk of intensive care unit admission/death was 83% lower (HR = 0.17, 95% CI: 0.08, 0.37) [323]. According to data from the UK based on 62,002 participants (age ≥ 16) across two periods: from 1 June 2021, to 27 November 27 2021, (Delta dominant (>70%), N = 4990) and from 20 December 2021, to 17 January 2022 (Omicron dominant (>70%), N = 4990), Omicron symptom duration was significantly shorter when compared to that of Delta (6.87 (95% CI: 6.58–7.16) days vs. 8.89 (95% CI: 8.61–9.17) days) [324] Omicron was also associated with a lower rate of hospitalization than Delta (1.9% vs. 2.6%, OR = 0.75 (0.57–0.98), *p* = 0.03) [324]. Similarly, when compared to 3075 Delta cases, a recent analysis of the first 1119 Omicron cases in France revealed much lower rates of hospitalization (1.9% for Omicron vs. 11.9% for Delta, *p* < 0.0001), intensive care (0.1% for Omicron vs. 3.1% for Delta, *p* < 0.0001), and mortality (0.1% for Omicron vs. 3.1% for Delta, *p* < 0.0001) [325].

Another study in Scotland based on national data collected between 1 November and 19 December 2021 suggests that Omicron is associated with a two-thirds reduction in the risk of COVID-19 hospitalization when compared to Delta [321]. According to a study conducted recently by the CDC (2022) COVID-19 incidence and hospitalization rates among unvaccinated people in Los Angeles County were 3.6 and 23.0 times those of fully vaccinated people with a booster, and 2.0 and 5.3 times those of fully vaccinated people without a booster, respectively, as of 8 January 2022, during Omicron predominance [326]. During the prevalence of Delta and Omicron, unvaccinated people had the highest incidence and hospitalization rates, whereas vaccinated people with a booster had the lowest rates.

According to the John Hopkins Bloomberg School of Public Health, hospital admissions in the United States were substantially higher than in Western Europe between December 2021 and January 2022 [327]. During that period, Americans were dying from COVID-19 at roughly twice the rate of individuals in the United Kingdom and four times the rate of Germans, which could be linked to herd immunity levels. The United States presently has the highest rate of COVID-19 mortality among wealthy countries [327]. Based on a South Korean report that documented the clinical presentation of Omicron cases, there were no severe incidences in the small sample size (n = 40) [328].

Lower hospitalization and mortality in Omicron cases could be related to protection induced by COVID-19 vaccines, booster doses, and previous SARS-CoV-2 infections, in addition to the newest variant being milder than other variants, such as Delta [296,329,330].

#### 2.6.4. Affinity to Angiotensin-Converting Enzyme 2 (ACE2) Receptors

According to Khatri et al. (2022), in the absence of TMPRSS2, Omicron may enter cells easily via ACE2-dependent endocytosis and is extremely prone to cathepsin L inhibitors, such as E64d [252]. Higher transmissibility is expected if the overlapping Omicron mutations maintain their known effects, especially because of mutations near the furin cleavage site [331]. Omicron has a 5.5 to 11 times greater mutation rate in the receptor-binding motif (RBM) than Alpha, Beta, and Delta SARS-CoV-2 variants [332]. The Q493R mutation in the RBD may enhance the affinity for hACE2 binding. In in vitro evolution experiments, the combination of mutations Q498R and N501Y dramatically improved the binding affinity to ACE2 [333]. The differences in the binding patterns of the wild-type and B.1.1.529 variant complexes revealed that key substitutions Asn417, Ser446, Arg493, and Arg498 in the B.1.1.529 RBD resulted in additional interactions with hACE2, whereas the loss of crucial residues in the B.1.1.529 NTD caused fewer interactions with three CDR regions (1–3) in the mAb [334].

Thus, the effect of Omicron on ACE2 is minimal, implying that the SARS-CoV-2 has already optimized its binding to ACE2 and that the virus has only a limited ability to improve its infectivity [335]. However, if the vaccination rate rises, variants may become increasingly destructive to vaccines in the coming years.

#### 2.6.5. Viral Load

The Omicron variant has two additional mutations in the nucleocapsid (N) protein, R203K and G204R, which are associated with elevated subgenomic RNA expression and viral loads [336,337]. Abdelnabi et al. investigated the infectivity of the Omicron variant in a hamster model in contrast to the ancestral D614G strain. As anticipated, in hamsters infected with the Omicron variant, the viral RNA content in the lungs was 3 log_10_ lower than in animals infected with D614G [338]. According to Hay et al., the Omicron variant had a lower peak viral load than the Delta variant. The mean peak Ct value for Omicron was 23.3 vs. 20.5 for Delta, underlining that the lower the Ct value, the higher the peak viral load [339]. When compared to individuals infected with the Delta variant, data collected in France during December 2021 on patients infected with the Omicron variant exhibited a lower viral load (higher Ct value) (22.7 for Delta vs. 24.4 for Omicron) [340]. Data from Switzerland also revealed that Omicron-infected individuals had lower infectious viral loads than Delta-infected patients (5.2-fold, 0.715 log_10_, *p*= 0.0185) [341]. Data collected from Geneva University Hospital found that in fully vaccinated patients, Omicron breakthrough infections resulted in a similar number of genome copies to Delta but much lower infectious VLs (14 times, 1.146 log_10_, *p* < 0.0001) [264]. Additionally, the same study showed that boosted individuals had a considerably decreased infectious VL but not RNA VL (5.3-fold, 0.7280 log_10_, *p* = 0.0004). According to a study conducted by Houhamdi et al. (2022) in France based on the first 1119 Omicron cases, the viral load of vaccinated individuals (mean Ct value = 23.2) was comparable to that of unvaccinated patients (mean Ct value = 23.4) [325].

Primary cases of patients infected with BA.2 who had not been vaccinated had a larger sample viral load (lower Ct value), with the median primary case having a 1.6 point lower sample Ct value [309].

#### 2.6.6. Reproduction Number (R_0_/R_t_)

Based on the findings of a study conducted in the UK, researchers estimated that Omicron’s reproduction number (R_t_) is higher than three based on specimens collected between 29 November and 11 December 2021 [319].

R_t_ was predicted to be 0.8–2.5 based on preliminary modeling of South African data (with the assumption that Omicron is driving the surge) [342]. Another study conducted by Gozzi et al. (2022) in South Africa showed that R_t_ was equal to 1.6 by 29 November 2021 [315]. Omicron’s R_t_ was estimated to be 4.2 times higher than that of the Delta variant (95% CI: 2.1, 9.1) [306]. A study conducted in South Korea by Kim et al. showed that the R_0_ was equal to 1.90 (95% CI: 1.50–2.43) during November and December 2021 [343]. Under the same epidemiological conditions, Ito et al. reported that the effective reproduction number (R_t_) of Omicron is 3.19 (95% CI: 2.82–3.61) times more than that of Delta in Denmark [344]. R_t_ was estimated at 2.5 in a study conducted in Florida, United States, during December 2021 [345]. Under the transmissibility conditions that existed in Italy in December, the reproduction number associated with the Omicron variant was in the range of 1.8–3.1 [346]. Furthermore, the R_t_ fluctuated between 1.34 and 3.57 depending on the location and vaccination rate, according to data obtained in many areas in India as of January 2022 [347]. According to recent research, Omicron has twice the reproduction number of the Delta variant, implying an R_0_ of 11.88 (95% CI: 9.16–14.61) [310,348]. Given the low vaccination coverage and high infectivity of Omicron, Bi et al. (2022) estimated that on 5 December 2021, the immunity-based effective reproduction number R_e_ of Omicron ranged from 7.0 to 9.4 throughout the 25 examined LMICs without any public health interventions [310].

#### 2.6.7. Vaccine Effectiveness and Vaccine Breakthrough

The RBD or N-terminal domain (NTD), which is crucial for binding and interacting with the ACE2 cellular receptor, contains the majority of the spike mutations linked to the decrease in antibody neutralization against Omicron compared to earlier variants [349].

Omicron is reported to be at least partially resistant COVID-19 vaccine-induced antibodies and to overcome major antibody-based immunotherapies [350,351,352]. As a result, a substantial number of breakthrough infections in vaccinated people have been documented around the world [353,354]. The poor cross reactivity of antibodies in suppressing Omicron replication, as well as replication in the URT, results in infection and transmission in vaccinated individuals [355].

According to research conducted by Discovery Ltd., South Africa’s largest private health insurer, Omicron lowered vaccine effectiveness against infection to 33%, down from 80% against Delta [295]. On the other hand, Collie et al. investigated the efficacy of the BNT162b2 vaccine against the Omicron Variant in South Africa, reporting a vaccine efficacy of 70% (95% CI: 62–76%) during the proxy Omicron period, which was supported by the results of all sensitivity tests [356].

Based on data collected until January 2022, a study conducted in Qatar found that mRNA booster immunization is associated with limited efficiency against symptomatic Omicron infection [304]. After 49 days of follow-up, the cumulative symptomatic infection incidence for BNT162b2 was determined to be 2.9% (95% CI: 2.8–3.1%) in the booster-dose cohort and 5.5% (95% CI: 5.3–5.7%) in the primary-series group of only two doses. In comparison to the primary series, booster effectiveness was 50.1% (95% CI: 47.3–52.8%). After 35 days of follow-up, the cumulative symptomatic infection incidence for mRNA-1273 in the booster-dose cohort was 1.9% (95% CI: 1.7–2.2%) and 3.5% (95% CI: 3.2–3.9%) in the primary-series cohort. In comparison to the primary series, booster effectiveness was 50.8% (95% CI: 43.4–57.3%) [304].

Although approved vaccines have been less effective in preventing infection, they provide considerable protection against Omicron, with booster doses (3X) providing a high level or protection, exceeding 80% under certain circumstances [317]. Based on 132 3D models of RBD antibody complexes, it was reported that Omicron may be twice as likely to evade existing vaccinations compared Delta [299]. Data gathered from 48 healthcare workers (HCWs) in the United States 3–4 weeks after receiving a second dosage of either Moderna mRNA-1273 (n = 20) or Pfizer/BioNTech BNT162b2 (n = 28) revealed that the Omicron had 22.9-fold (*p* < 0.001) higher neutralization resistance than the ancestral D614G [183]. The sensitivity of 28 serum samples from COVID-19 convalescent patients infected with the original SARS-CoV-2 strain was tested against pseudotyped Omicron, as well as other VOCs (Alpha, Beta, Gamma, and Delta) and VOIs (Lambda and Mu) [299]. According to data from the state of New York, USA (n = 8,834,604), between May and September 2021, Rosenberg et al. (2022) reported high vaccine efficacy in preventing severe disease in people over the age of 65, with varying levels of protection conferred by different vaccines—95% for BNT162b2, 97% for mRNA-1273, and 86% for Ad26.COV2.S—and minimal declines in protection 6 months after vaccination [357].

According to the results of in vitro experimentation, the Omicron variant may result in a greater escape from immune protection generated by previous SARS-CoV-2 infection and possibly even current COVID-19 vaccines. According to a recent study by Ai et al., the Omicron variant had lower neutralizing sensitivity to immune sera elicited by vaccines after boost than other SARS-CoV-2 variants [358]. Despite a significant drop in pVNT against Omicron (5.86–14.98 times) compared to the prototype, 100% of samples demonstrated good neutralization activity against Omicron 14 days after homologous or heterologous vaccine boosters. Their findings suggest that Omicron is more likely than prototypes and other VOCs to evade vaccine-induced immune protection. Pearson et al. (2021) reported a three- to eightfold decrease in neutralization titers for Omicron compared to Delta in a study conducted in South Africa [359]. Moreover, according to the UK Health Security Agency, an estimated 20- to 40-fold reduction in neutralizing activity by two doses of BNT162b2 [360] and a reduction of more than 10-fold for both ChAdOx1 and mRNA-1273 vaccines was reported [361,362].

When compared to 25 weeks after the second vaccine dose, a study in Scotland indicated that third/booster vaccine doses of BNT162b2 and mRNA-1273 (known as Pfizer-BioNTech and Moderna, respectively) provided significant extra protection against symptomatic disease [321]. When compared to Delta, Omicron appears to be associated with a two-thirds lower risk of COVID-19 hospitalization. Whereas the second vaccine dose provides the best protection against Delta, the third/booster dose was associated with a 57% (95% CI: 55, 60) reduction in the risk of symptomatic infection for Omicron when compared to 25 weeks after the second dose [321]. Research conducted in Korea revealed that 60% of Omicron VOC-positive patients were unvaccinated, emphasizing the importance of immunization in limiting the spread of the disease [363].

To conclude, the quickest and most widely circulating variant is Omicron. Omicron appears to prefer infecting and multiplying in the upper respiratory tract compared to Delta and other strains. When compared to Delta, the viral load was found to be lower. Depending on the region and vaccination rate, the reproduction number of Omicron ranges from 1.34 to less than 4. Despite the fact that licensed vaccinations are less effective at preventing infection, they still offer significant protection against Omicron, with booster doses (3×) providing significantly more protection, especially given the lower observed hospitalization and mortality rates.

## 3. Epidemiological Factors

### 3.1. Weather Conditions and Environmental Factors

As COVID-19 spreads across the globe over the course of several seasons, the role of weather and environment in transmission becomes increasingly important. The transmission mechanism of COVID-19, like that of other respiratory viruses, is thought to be linked to local weather conditions, including but not limited to air pollution, wind speed, temperature, and humidity [364].

#### 3.1.1. Air Pollution

According to data obtained in fifty-five Italian cities from March to May 2020, during the first wave of the COVID-19 pandemic, the spread of COVID-19 in cities with high levels of air pollution resulted in a higher number of COVID-19-related infected individuals and deaths. The number of affected individuals was higher in cities with more than 100 days per year surpassing PM_10_ or ozone restrictions, cities situated in hinterland zones (i.e., away from the coast), cities with a low average wind speed, and cities with lower average temperatures, according to the findings [365,366]. Furthermore, the first wave of the COVID-19 pandemic in Italy resulted in more than 75% of infected and 81% of deaths in industrialized areas with high levels of air pollution, according to the findings [365]. Thus, air pollution can boost the lethality COVID-19 and delay its recovery [367].

Exposure to high levels of PM_2.5_ was associated with a 10% (95% CI: 8–12%) increased risk of hospitalization among COVID-19-infected patients, according to data gathered in the United States between 2 March 2020 and 31 January 2021 [368]. At PM_2.5_ levels below the permissible standards (12 g/m^3^), the danger was obvious. In the absence of UV light, viral sensitivity to temperature is at its peak [369].

Severe air pollution is significantly linked with increased confirmed cases and mortality associated with COVID-19, according to data gathered in Pingtung County, Southern Taiwan, between 14 June and 11 July 2021 during the peak of Delta variant circulation. For example, air pollutants such as PM_2.5_, PM_10_, and NO_2_, as well as high wind speeds, were clearly and positively correlated with the COVID-19 spread—specifically the Delta variant (e.g., increased morbidity and mortality of COVID-19 cases) [370].

#### 3.1.2. Temperature, Humidity, and Wind Speed

After controlling for population migration, a multicity study in China revealed that meteorological conditions have an independent impact in on COVID-19 transmission. Low temperatures, mild diurnal temperature variation, and low humidity are all likely to promote transmission [371]. Haque and Rahman (2020) used a linear regression framework to show that high temperature and humidity considerably reduce COVID-19 transmission in Bangladesh from March to May 2020 [372]. Using data from 100 Chinese cities and US counties from January to April 2020, Wang et al. (2021) observed that the effective reproduction number of COVID-19 decreases with increasing air temperature and relative humidity [373]. Khan strongly suggested that higher temperatures may alter (i.e., aggravate) R_0_ values and that regions with older populations are more likely to be affected [88]. A study conducted by Xie and Zhu on the relationship between ambient temperature and COVID-19 infection in 122 Chinese cities revealed that a 1 °C increase in temperature was associated with a 4% increase in daily COVID-19 cases when the mean temperature (lag 0–14) was less than 3 °C [374]. In a systematic review and meta-analysis of 11 studies, temperature (0.22 (95% CI: 0.16–0.28)), humidity (0.14 (95% CI: 0.07–0.20)), and wind speed (0.58 (95% CI: 0.49–0.66)) were found to be significantly correlated with COVID-19 [375].

On the other hand, high temperature, relative humidity, and solar radiation were found to have a negative relationship with coronavirus spread, particularly the Delta variant [370]. According to Zoran et al. (2022), favorable atmospheric stability conditions in Madrid, Spain, from January 2020 to July 2021 aided COVID-19 disease spread [376].

A recent metaregression of 158 studies conducted by Tan et al. (2022) revealed that Asian countries had more positive associations between COVID-19 spread and rising temperatures compared to other regions. In addition, regardless of the statistical analytic method or geographic location, increased sun energy was linked to reduced COVID-19 spread [377].

In contrast, according to a systematic review by Ansar et al. (2021) of 23 studies from the United States, Europe, Asia, and the Americas, no direct association between temperature and humidity and the global spread of COVID-19 infections was established in other investigations [378]. However, the same study found that a lower air quality index combined with a higher wind speed may boost COVID-19 transmission in specific circumstances [378].

### 3.2. Indoor Settings

The majority of SARS-CoV-2 clusters have been attributed to indoor settings, implying that virus stability under indoor environmental conditions could be a key factor in superspreading risk [379].

After the opening of all UK schools in September and October 2020, adult and child cases spiked, requiring a second national lockdown for adults from 5 November to 2 December 2020. Despite the fact that all schools were open at the time, the number of cases declined quickly, first among adults and then among children [380,381]. A study conducted by Lam-Hine et al. based on data collected between May and June 2021 showed that despite masking, the attack rate in the two rows closest to the teacher’s desk was 80% (8/10), and the attack rate in the three back rows was 28% (4/14) [382]. Before receiving a test, the sick teacher, who was not vaccinated, continued to work for two days. Despite school guidelines to mask when indoors, the teacher read aloud to the class on occasion during this period, leading to a total of 26 Delta infections [382]. Other studies also found that the Delta variant is particularly transmissible in indoor environments [382,383].

A total of 47 COVID-19 cases (all 21 tested cases were Delta) were linked to a gymnastics facility from 15 April to 3 May 2021. The overall attack rates for facilities and households were 20% and 53%, respectively. Four people (9%) had received two doses of the Moderna or Pfizer-BioNTech vaccine or a single dose of the Johnson & Johnson vaccination 14 days prior to a positive test result. Several potential transmission risk factors were identified, including a lack of mask use among active participants, as well as increased respiration during active sport participation (furthermore, facility policy required that all persons not actively participating wear masks, but this policy was not always followed) and poor facility ventilation [383].

According to WHO, COVID-19 is more likely to be contracted in the following situations, which should be avoided: closed places with insufficient ventilation, crowded areas with a large number of individuals present, and close contact with others, such as close-range conversations [296].

### 3.3. Population Density, Race and Life Condition

Population density is another important factor in COVID-19 transmission. A study based on data collected during the period from January to 1 September 2020 in the US showed that the most important factors for estimating daily COVID-19 cases are the population and population density, followed by the social distance index and weather conditions [364]. Bozzuto et al. (2021) found that countries with low-to-medium population size never produced high corrected R_0_ values, implying that the pace of dissemination of COVID-19 is restricted by population density [384]. Additionally, Ilardi et al. (2021) reported on a substantial positive linear association between population density and the number of cases, deaths, and case-fatality rate [385]. Another study using data from 34 European countries found that a higher proportion of the population residing in cities, a lack of rapid mobility reduction, and delayed government restrictions were all linked to a higher COVID-19 peak death rate [386].

Results based on data collected in Malaysia between 19 January and 31 December 2020 showed that districts with high population densities and strong links to other districts, whether physically, socioeconomically, or in terms of infrastructure, tend to have COVID-19 cases surges within weeks of one other [387]. Population density was found to have a fairly significant association with cumulative COVID-19 cases (*p*-value of 0.000 and R2 of 0.415) and a weak relationship with COVID-19 infection rates based on a parametric method using Pearson correlation (*p*-value of 0.005 and R2 of 0.047) [387]. Based on data collected between 2 March and 31 May 2020, a study conducted by Chishinga et al. (2020) in Georgia, United States, revealed that non-Hispanic Blacks (AOR 1.9, 95% CI: 1.5–2.4) and Hispanics (AOR 1.7, 95% CI: 1.2–2.5) had higher risks of severe COVID-19 than non-Hispanic Whites [388].

On the other hand, other studies showed poor or no association with population density [389,390,391].

When compared to White people in the COVID-19 study, those of the Black race and those who live in impoverished neighborhoods were identified as population categories that may be more vulnerable to the unfavorable influence of PM_2.5_ on hospitalization risk [368]. In a study conducted in South Africa by Vermeulen et al. (2021) (data from 16,762 test blood donations received between January and May 2021 for anti-SARS-CoV-2 antibodies) seroprevalence in Black donors was consistently several times higher than in White donors, with the other major population groups (Colored and Asian) not well represented in any regions [160].

Non-Hispanic White people in the United States, on the other hand, accounted for fewer admissions among fully vaccinated adults during Omicron dominance than during Delta dominance (46.6% vs. 62.4%; *p* = 0.01) [392].

### 3.4. Age

#### 3.4.1. Children

The prognosis of COVID-19 patients has been demonstrated to be strongly influenced by their age. The G614 variant was not related to higher levels of hospitalization, although older age [393,394], male sex [394,395], and higher Ct values (lower viral loads) were strongly predictive of hospitalization, according to a regression analysis [69]. 614G has been linked to a younger patient age, according to Volz et al. (2021) [71]. Early in the pandemic, children accounted for 1–2% of diagnosed COVID-19 infections, according to the bulk of published studies [396]. Various reports revealed that the median age of the diagnosed youngsters ranged from 3.3 to 11 years [396,397]. Children were typically asymptomatic or had milder symptoms than adults after exposure to infected family members [398,399]. When compared to infected adults and older children, younger children with mild/moderate COVID-19 due to the original variant were more likely to have higher nasal viral loads, allowing for a silent spread of the virus, particularly with the sequential opening of schools, which could be a precursor to second and third waves [400].

A systematic review conducted by Biswas et al. (2021) revealed that children < 10 years of age had 0.35-fold lower odds of being infected than those aged 30–39. There were large differences in the odds of infection according to nationality, with Bangladeshi and Nepalese individuals having the lowest odds of infection [401].

A study assessing the transmissibility of SARS-CoV-2 lineage B.1.1.7 in England found that more infection cases were observed among patients younger than 20 years of age [97]. Children are more likely than adults to be asymptomatic or have a moderate, temporary, and self-limiting upper respiratory tract infection after being exposed to the virus.

Due to the more transmissible nature of the Delta variant, the full resumption of in-person schooling, as the unvaccinated status of practically every American under the age of 12, many children in the United States contracted COVID-19 in the summer of 2021. According to the American Academy of Pediatrics, children under the age of 18 accounted for 28.9% of reported COVID cases in the first week of September 2021 [402]. Younger people and non-Hispanic Black people were more likely to be infected with the Delta variant [403]. A study conducted by Hwang et al. (2021) in South Korea showed that about half of affected people were under the age of 19 [404]. Kumar et al. (2021) found that in comparison to the wild-type (B.1) strain, the Delta variant affected a higher proportion of younger age groups, particularly those younger than 20 years (0–9 years: 4.47% vs. 2.3%, 10–19 year: 9% vs. 7%) [405].

In comparison to rates during the Delta-dominant period, age-adjusted incidence and hospitalization rates increased in all groups, regardless of vaccination status, as Omicron became dominant, although it was less likely to cause serious sickness [406]. Children have been no exception, accounting for a higher proportion of COVID-19 hospitalizations in the United States than at any previous time throughout the pandemic [407]. Children under the age of two accounted for around 10% of all hospital admissions owing to Omicron infection in Tshwane, South Africa, according to the National Institute for Communicable Diseases of South Africa [408]. Preliminary subanalyses found that Omicron cases in school-aged children (5 to 17 years old) had a reduced probability of hospitalization compared to Delta cases in the same age range (HR 0.42, 95% CI: 0.28–0.63) [360].

Children under the age of five years account for 2% of reported global cases and 0.1% of reported global deaths according to age-disaggregated data reported by the WHO from 30 December 2019 to 25 October 2021 [409]. Children and adolescents aged 5 to 14 years account for 7% of all reported cases and 0.1% of all recorded fatalities worldwide [409].

#### 3.4.2. Adults and Older Age

Adults, on the other hand, are more likely to develop severe disease, require hospitalization, and die from COVID-19 [410,411,412]. In the literature, older age has been identified as the most important risk factor for severe COVID-19 disease [413].

Early in the pandemic, elderly people infected with the SARS-CoV-2 virus were more likely to develop severe infection and sequelae, resulting in higher rates of morbidity and fatality. Case fatality was 8.0% in patients aged 70–79 years and 14.8% in patients aged 80 years and older, according to a comprehensive study encompassing 72,314 patients in China [414]. In a retrospective cohort study by Zhou et al., which included 191 hospitalized COVID-19 patients in Wuhan, multivariable regression revealed that older age was associated with an increased risk of in-hospital death (odds ratio 1.10, 95% CI: 1.03–1.17, per year increase; *p* = 0.0043) [415]. According to an analysis of the case-fatality rate in the United States, the estimated overall death rate ranged from 0.4/1000 for those under the age of 18 to 304.9/1000 for those over the age of 85 [416]. Another study from Europe revealed 48% mortality in patients aged ≥85 years [417]. Additionally, a systematic review and meta-analysis conducted by Du et al. (2021) revealed that older age was significantly associated with severe illness (OR  =  2.62, 95% CI : 2.01–3.42) and death (OR  =  6.00, 95% CI : 3.48–10.34) [418]. In another systematic review, Biswas et al. (2021) found that patients older than 50 years of age had a 15.4-fold greater risk of mortality compared to those under the age of 50 (RR 15.44, 95% CI: 13.02–18.31; *p* < 0.00001) [401].

Furthermore, between November 2020 and February 2021, people aged 85 years and older in England had the highest risk of dying (6.87%, CI: 4.33–9.42) within 28 days of a SARS-CoV-2-positive test due to B.1.1.7, especially those with two or more comorbidities [419]. A study conducted in Spain showed that patients older than 65 carrying the B.1.1.7 variant have a twofold higher chance of being admitted to hospital (OR 2.05; 95% CI: 1.32 3.19) [112].

Gamma cases reported by seven European countries had a higher mean age than non-VOC patients (46 years vs. 40 years; *p* < 0.05) [208]. In the second wave (December 2020 to February 2021), a progressive increase in the percentage of patients aged 60 years or older admitted to the ICU was observed, in contrast to the demographic pattern observed during the first wave (May to July 2020). Moreover, during the second wave, a higher number of younger patients required critical care. More COVID-19 patients with no comorbidities were admitted to the ICU in February 2021 [211]. After the emergence of the P.1 variant, a study in southern Brazil reported that individuals aged 40 to 59 years were at a 7.7 times higher risk of death (95% CI: 5.01–11.83; *p* < 0.0001) [209].

In China, older age was found to be an independent risk factor for severe Delta variant cases, especially in those older than 58.5 years, with each 1-year increase in age (OR, 1.089; 95% CI: 1.035–1.147; *p* = 0.001) representing a risk factor for severe cases (OR, 1.089; 95% CI: 1.035–1.147; *p* = 0.001). Furthermore, in patients older than 58.5 years, the probability of progressing to severe instances increased 13.444-fold [420]. In addition, according to data collected between 4 April and 17 July 2021, during the spread of the Delta variant in the United States, older age groups (≥65) had higher hospitalization and death rates, independent of vaccination status [421]. In India, patients aged 60 and older were approximately four times more likely to be admitted to hospital as a result of infection with the Delta variant [422].

Moreover, increased immunization was linked to a decreased risk of ICU admission (*p* = 0.02) and, among people 65 years and older, a lower likelihood of death, compared to hospitalizations (*p* = 0.04) among adults during Omicron predominance in the United States [392,423]. 

### 3.5. Gender

During the pandemic, men and women were affected differently. A major retrospective study on COVID-19 patients admitted to the ICU in the Lombardy, Italy, revealed that 82% of patients were men [424]. Men were reported to account for 62% of hospitalized patients in China [415]. After controlling for other risk variables, several other studies indicated that male COVID-19 patients had a higher chance (adjusted odds ratio 1.4, 95% CI: 1.1–1.6) of severe outcome and/or mortality than female COVID-19 patients [388,425,426].

Based on a meta-analysis of 20 studies, males were found to have a significantly higher risk of mortality than females (RR = 1.86; 95% CI: 1.67–2.07) [401]. However, women were found to have 1.29-fold higher odds of infection than men.

In a study conducted in Brazil after the emergence of the P.1 variant, females aged 20 to 39 years with no pre-existing risk conditions had a 5.65 times higher risk of death in February 2021 (95% CI: 2.9–11.03; *p* < 0.0001) [209].

In India, the proportion of women who became infected with the Delta variant increased (41% vs. 36%), as did the risk of mortality (odds ratio: 3.034, 95% CI: 1.7–5.2, *p* < 0.001), with a higher proportion of women dying compared to men (32% vs. 25%) [405].

Based on data collected between 1 March 2020 and 30 November 2020, a large analysis of 308,010 adults with COVID-19 hospitalized at US academic centers showed that men have a higher rate of respiratory intubation than women (21.4% vs. 14.6%, *p* < 0.001) and longer hospital stays (9.5 ± 12.5 days vs. 7.8 ± 9.8 days, *p* < 0.001), as well as a higher death rate (13.8% vs. 10.2%, *p* < 0.001), even when compared across age groups, race, and comorbidities [427].

Last but not least, Omicron was also found to affect men and women differently. According to a survey conducted by the American platform Web MD, women infected with the Omicron variant of COVID-19 experience higher levels of fatigue. Despite the fact that fatigue is a typical symptom among individuals infected with Omicron, roughly one-third of men stated they had experienced it, whereas nearly 40% of female participants reported excessive exhaustion [428].

## 4. COVID-19 Disease Clinical Treatment Methods

Early in the pandemic, knowledge of COVID-19 and its therapeutic treatment was poor, prompting a rush to develop therapeutics and drugs to combat this novel viral infection. Since then, due to the relentless work of clinical researchers around the world, substantial progress has been made, resulting in improved understanding of not only COVID-19 and its treatment but also the rapid development of innovative therapies.

In addition to oxygenation and ventilation therapy, antiviral therapies, anti-SARS-CoV-2 neutralizing antibodies, anti-inflammatory therapies, and immunomodulatory agents are available to treat COVID-19 disease [429]. The clinical utility of these treatments is dependent on the severity of the illness or certain risk factors. COVID-19 illness includes two phases; the first occurs before or shortly after the onset of symptoms and is characterized by high SARS-CoV-2 replication [430]. During this stage of viral replication, antiviral medications and antibody-based treatments are expected to be more effective. A hyperinflammatory condition generated by the production of cytokines and the activation of the coagulation system generates a prothrombotic state in the later stages of the illness. Anti-inflammatory medications such as corticosteroids, immunomodulating therapies, or a combination of these therapies may be more effective than antiviral therapies in combating this hyperinflammatory state [431]. The following is a list of the most recent potential therapeutic approaches that have been proposed or approved.

### 4.1. Oxygenation and Ventilation

Patients with COVID-19 who have respiratory insufficiency should be closely monitored with continuous pulse oximetry [430]. Maintaining a respiratory frequency (RF) of 24/min, an oxygen saturation (SpO_2_) between 92 and 96%, and/or a pO_2_ of 8.5 kPa is the primary focus of treatment choices [432]. If the patient does not maintain these levels, oxygen therapy is started and gradually increased depending on the severity of the respiratory hypoxia.

### 4.2. Antiviral Therapies

Patients over the age of 12 who have COVID-19 pneumonia and require oxygen therapy but are not mechanically ventilated (SATO_2_ 94% without oxygen therapy or subjective requirement for oxygen therapy and lung infiltrations) should be administered antiviral medications [432]. Remdesivir, molnupiravir, hydroxychloroquine, and chloroquine are examples of medications used to treat COVID-19 disease.

Initially, antiviral therapies for COVID-19 were considered, including hydroxychloroquine and chloroquine. Results of randomized control trials comparing the use of hydroxychloroquine with or without azithromycin in hospitalized patients to placebo showed no improvement in clinical status or overall mortality [433,434]. The use of hydroxychloroquine as a postexposure prophylactic did not prevent SARS-CoV-2 infection or symptomatic COVID-19 sickness in randomized, controlled trials [435,436].

Three randomized, controlled clinical trials revealed that remdesivir was superior to placebo in reducing time to recovery and mortality in adults hospitalized with mild-to-severe COVID-19 [437,438,439]. Only a group of patients with COVID-19 pneumonia who required oxygen therapy had lower mortality [437]. Remdesivir has been approved by the US Food and Drug Administration (FDA) for use in adults and pediatric patients (over the age of 12 and weighing at least 40 kg) who are hospitalized with COVID-19 [430]. In a randomized, double-blind, placebo-controlled experiment, Gottlieb et al. (2022) reported that when at-risk, non-hospitalized COVID-19 patients were treated with a 3-day course of remdesivir, they had an 87% lower chance of hospitalization or mortality than patients who received the placebo [440]. There is no evidence that remdesivir is effective against the novel SARS-CoV-2 variants.

Molnupiravir is an antiviral medication that works by inhibiting the RdRp enzyme [430]. It was first created as a therapy for influenza and alphaviruses. Molnupiravir was found to result in a significant reduction in hospitalization and death in cases of mild COVID-19 disease based on a meta-analysis of available phase 1–3 studies [441]. Early treatment with molnupiravir reduced the risk of hospitalization or mortality in at-risk unvaccinated people with mild-to-moderate, laboratory-confirmed COVID-19, according to findings from a phase 3 double-blind, randomized, placebo-controlled trial [269].

### 4.3. Anti-SARS-CoV-2 Neutralizing Antibodies

Patients who recover from COVID-19 produce neutralizing antibodies against SARS-CoV-2, although the titer of these antibodies in the blood diminishes significantly six months after infection. Nonetheless, their role as therapeutic agents in the treatment of COVID-19 has been investigated in a number of clinical trials [430]. If administered early during the infection course, convalescent plasma therapy may minimize the incidence of serious COVID-19 pneumonia. Convalescent plasma treatment was approved by the FDA for patients with severe life-threatening COVID-19 [442]. Multiple trials assessing the use of convalescent plasma in life-threatening COVID-19 have had inconsistent outcomes, despite the fact that it appeared to be a promising treatment option. According to one retrospective study based on a US national registry, patients hospitalized with COVID-19 who received a transfusion of convalescent plasma with higher anti-SARS-CoV-2 IgG antibody levels had a lower risk of death than patients who received a transfusion of convalescent plasma with low antibody levels. Patients treated with convalescent plasma vs. standard therapy showed no significant differences in clinical improvement or overall mortality in three small randomized, controlled trials [443,444,445]. Convalescent plasma may be used in patients with severe immune deficiencies [432].

Additionally, REGN-COV2 (casirivimab and imdevimab) is a cocktail of two non-competing IgG1 antibodies that target the RBD on the SARS-CoV-2 spike protein and have been proven to reduce viral load in vivo, avoiding virus-induced pathological sequelae. In non-hospitalized individuals with COVID-19, preliminary findings from a phase 3 trial of REGN-COV demonstrated a 70% reduction in hospitalization or death [430]. Weinreich et al. (2021) reported that the REGN-COV2 antibody cocktail reduced viral load relative to placebo in 275 patients in an ongoing double-blind trial involving non-hospitalized COVID-19-positive individuals assigned to receive placebo, 2.4 g of REGN-COV2, or 8 g of REGN-COV2 [446].

Sotrovimab (VIR-7831) is a potential neutralizing monoclonal antibody that has shown in vitro effectiveness against all four VOCs: Alpha, Beta, Gamma, and Delta. Gupta et al. found that one dose of sotrovimab (500 mg) reduced the risk of hospitalization or mortality by 85% in high-risk, non-hospitalized patients with mild-to-moderate COVID-19 compared to the placebo in the double-blind, placebo-controlled phase 3 COMET-ICE trial [447].

It is worth noting that the FDA approved REGN-COV2 and sotrovimab for clinical use in non-hospitalized patients (aged 12 years and older and weighing at least 40 kg) with laboratory-confirmed SARS-CoV-2 infection with mild-to-moderate COVID-19 and who are at high risk of advancing to severe disease and/or hospitalization [430].

### 4.4. Steroid Treatment

The efficacy of glucocorticoids in COVID-19 patients was not well described during the early stages of the pandemic. Dexamethasone has been found to reduce mortality in oxygen-dependent patients [448]. In patients who do not require oxygen therapy, no effect has been observed. The Randomized Evaluation of COVID-19 Therapy trial found that using dexamethasone reduced 28-day mortality among patients on invasive mechanical ventilator or oxygen support but not in those who were not [449]. In hospitalized patients who require supplementary oxygen or non-invasive or invasive mechanical breathing, dexamethasone is currently regarded the standard of therapy, either alone or in combination with remdesivir, depending on the severity of sickness [430].

### 4.5. Immunomodulatory Agents

For the treatment of COVID-19, randomized studies of the interleukin-6 (IL-6) antagonist tocilizumab have yielded mixed results. Some RCTs, such as COVACTA (A Study to Evaluate the Safety and Efficacy of Tocilizumab in Patients with Severe COVID-19 Pneumonia), revealed no change in mortality on day 28, but several studies reported positive results, with reduced chance of requiring intensive therapy or death [450,451,452]. In the REMAP-CAP and RECOVERY pragmatic platform studies, IL-6 antagonist treatment resulted in marginally decreased mortality (REMAP-CAP, hospitalization mortality: 27% vs. 36%; RECOVERY, 28-day mortality: 31% vs. 35%) [453,454]. Results from several studies showed that interferon-β-1a (IFN-β-1a) and interleukin (IL)-1 antagonist therapies (e.g., anakinra) are not recommended to treat COVID-19 infection, owing to a lack of data [455,456,457,458]. Interferons are cytokines that are required to mount an immune response to a viral infection, and SARS-CoV-2 inhibits their release in vitro [459].

## 5. Transmission Rate of Other Coronaviruses

There are seven CoVs reported to infect humans, including three that cause severe diseases: SARS-CoV-2, SARS-CoV, and MERS-CoV [4,460]. Several seasonal coronaviruses, including the alphacoronaviruses HCoV-NL63 and HCoV-229E, as well as the betacoronaviruses HCoV-OC43 and HCoV-HKU1, are known to circulate in the human population and normally cause relatively mild respiratory tract infections, mostly in children (15–30%) [461,462].

COVID-19 is suspected to have initially been transmitted from animal to human at a seafood market in China and gained the ability to transmit among humans, causing the largest pandemic of a coronavirus to date [463]. This SARS-CoV-2 differs from other betacoronaviruses in that it has a unique combination of polybasic cleavage sites, a trait that has been shown to boost pathogenicity and transmissibility in other viruses [464].

SARS-CoV was an extremely fatal virus that faded away following extensive public health interventions [465]. In contrast, the novel SARS-CoV-2 virus, which emerged in December 2019, quickly spread throughout the world, causing a global pandemic. The 2003 SARS outbreak ended in June 2003, with a total of 8098 cases reported worldwide, 774 deaths, and a case fatality rate of 9.7%, with the majority of cases acquired nosocomially [465]. MERS-CoV, on the other hand, emerged in 2012 and has caused 2494 recorded cases and 858 deaths in 27 countries, with a case fatality rate of 36% [466,467]. Saudi Arabia had the largest number of MERS cases (n > 1700) in the world [467]. The MERS-CoV outbreak in Jeddah, Saudi Arabia in 2014 revealed a higher risk of secondary human-to-human transmission in healthcare settings when the majority of patients had contact with a healthcare facility, other patients, or both [468]. MERS-CoV has been a hospital epidemic, causing MERS in medical workers [469,470]. Individuals who have been in direct contact with proven cases prior to or after symptom development, whether at work, at home, or in the hospital (medical staff), are classified as high-risk [467]. In addition, between May and June 2015, South Korea witnessed the largest MERS-CoV outbreak, with a case fatality risk of 21% (95% CI: 14–31). A 68-year-old man returned from Saudi Arabia, where he contracted MERS, and disseminated the virus across South Korean hospitals after repeated hospitalizations, causing an outbreak in South Korea [471,472]. Nearby immunodeficient individuals, sharing of healthcare tools and workers, and poor environmental cleanliness are all factors that can influence how a “superspreader” is interpreted [473]. In a study conducted by Kang et al. (2017) superspreaders were detected in around 33.3% of laboratory-confirmed infected hospital patients [474]. In Jordan in 2015, a variant virus with unique deletions in the accessory open reading frames caused a MERS-CoV outbreak [475]. Despite the accumulation of mutations, the MERS-CoV deletion variants remained human-to-human transmissible and caused clinical disease in infected people.

MERS-CoV-specific antibodies, as well as viral shedding of related viruses, has been discovered in dromedary camels (Camelus dromedarius). As a result, dromedaries are thought to be the primary source of MERS-CoV transmission to humans, despite the fact that the origin of the virus is still unclear [476,477,478]. Additionally, droplet transmission and close-contact transmission are two probable MERS-CoV transmission modes between people [479]. SARS-CoV is transmitted from person to person mostly by droplets, and the population is typically susceptible; SARS-CoV-2 spreads by droplets and contact among the population. SARS-CoV-2 may be transmitted through aerosols in high-concentration situations [479].

Unlike SARS-CoV, which evolved from animals and was eradicated from the intermediate host reservoir, MERS-CoV is common in dromedary camels, resulting in sporadic zoonotic infections. SARS-CoV-2 is less lethal (1–3%) than MERS-CoV and SARS-CoV but significantly more transmissible, the R_0_ for SARS-CoV-2 being the highest [480]. Before public health control measures were implemented, the R_0_ for the SARS outbreak of 2003 was anticipated to be between 2.0 and 3.0 in the early months (until the end of April) [465,481,482]. Various control techniques soon lowered the transmissibility to 1.1 (IQR: 0.4% to 2.4%) [482,483]. The R_0_ for MERS-CoV was calculated at 0.69 (95% CI: 0.50–0.92), indicating that the virus never created a prolonged epidemic [484,485]. Furthermore, according to the literature, the range of R_0_ values for MERS without control was 0.8–1.3 [480,485]. COVID-19, like SARS, has an R_0_ greater than 1 and is extremely contagious [486]. SARS-CoV-2 has been found to have higher rates of transmissibility and pandemic risk than SARS-CoV, with an effective reproduction number (R_t_) ranging from 1 to 7, depending on the variant [460,483].

The incubation period is another crucial aspect that influences virus transmission. The incubation periods of SARS, MERS, and COVID-19 are all comparable [479]. COVID-19 incubation times have been reported to range from 2.87 days to 17.6 days [47,480,487,488]. As for MERS, a systematic review based on 59 studies found that the incubation period in Saudi Arabia was 4.5–5.2 days, whereas in South Korea, it was 6–7.8 days [489]. Additionally, the incubation period for SARS-CoV was discovered to be between two and seven days [490,491,492].

On the other hand, the kinetics of virus shedding differ significantly between SARS-CoV, SARS-CoV-2, and MERS-CoV [480]. SARS-CoV and MERS-CoV have reduced airway tropism, with less virus in the upper respiratory tract, but SARS-CoV-2 has a distinct tropism. Within the first five days of SARS-CoV-2 infection, the average viral load per upper respiratory tract sample in a family cluster was 6.8 × 10^5^ copies per swab, and live virus isolates were isolated from swabs during the first week of sickness [493]. Researchers discovered a high viral load during the outset of symptoms, which decreased over the next 5–6 days [494]. This rapid drop in viral load renders isolation and quarantine of SARS-CoV-2 patients and their contacts much more difficult and ineffective, as it must be done as quickly as possible after onset of sickness to prevent transmission. SARS-CoV virus levels, on the other hand, peaked 6–11 days after symptom onset, giving researchers an extra week to discover and isolate individuals before transmission [495,496]. This distinction could explain why, unlike the SARS-CoV-2 pandemic, SARS was able to be eradicated in 2003.

Furthermore, Feikin et al. investigated whether Ct levels were associated with clinical severity in Saudi Arabian patients whose nasopharyngeal specimens were positive for this virus as determined by real-time reverse-transcription PCR [497]. The median Ct for the upstream of the E-gene target was 31.0 in 102 patients, which is considerably lower than the median of 33.0 among 61 survivors (*p* = 0.0087) [497]. Another study conducted in 2015 in South Korea revealed that the number of copies of MERS-CoV RNA detected by rRT-PCR in respiratory samples peaked during week 2, with a median value of 7.21 log_10_ copies per milliliter in the severe group and 5.54 log_10_ copies per milliliter in the mild group (*p* = 0.06) [498]. In the severe group, the peak viral load in sputum or tracheal aspirate was larger and occurred later than in the mild group [498].

To conclude, MERS-CoV had the highest fatality rate in comparison to SARS-CoV and SAR-CoV-2. MERS-CoV has presented as hospital outbreaks, and dromedaries are thought to be the main source of human MERS-CoV infection. MERS-CoV, SARS-CoV, and SAR-CoV-2 spread by droplets and human contact. SARS-CoV-2 is less fatal than MERS-CoV and SARS-CoV, but it is much more transmissible, with R_0_ being the highest for SARS-CoV-2. SARS, MERS, and COVID-19 all have similar incubation durations. SARS-CoV and MERS-CoV have a lower airway tropism, with less virus in the upper respiratory tract, but SARS-CoV-2 has a unique tropism.

## Figures and Tables

**Figure 1 vaccines-10-00919-f001:**
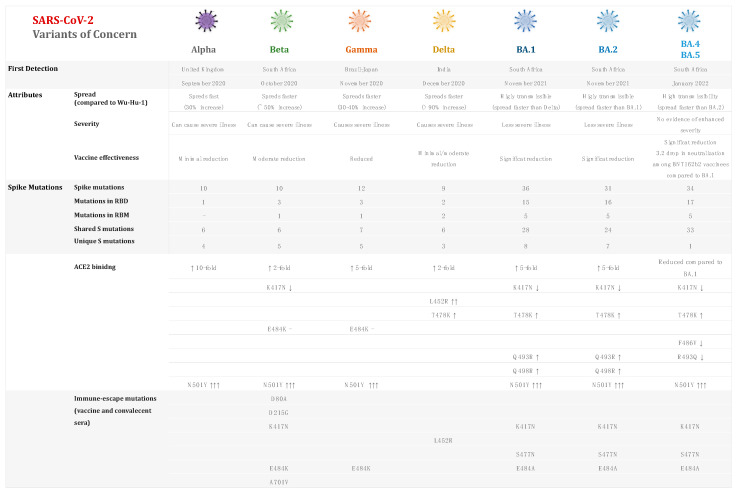
COVID-19 variants of concern (VOC), their origin, and characteristics. Arrows represent relative increase or decrease in the binding affinity to ACE2 receptor.

**Figure 2 vaccines-10-00919-f002:**
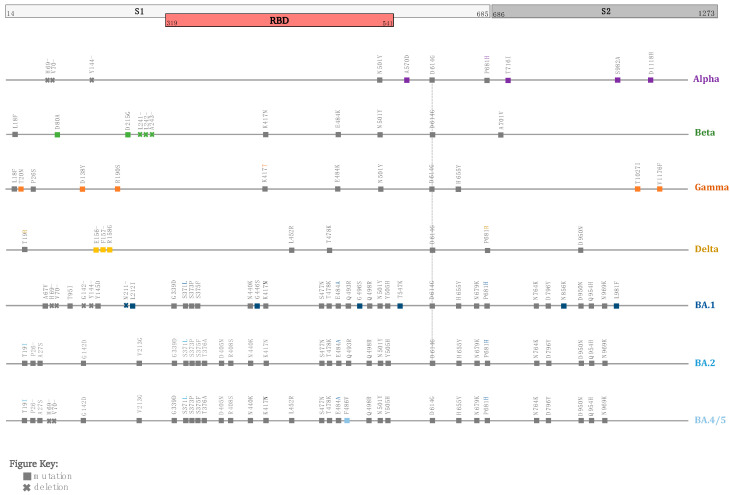
Signature mutations in different SARS-CoV-2 circulating VOCs. Lineage-specific mutations are colored, and shared mutation are indicated in gray.

## Data Availability

Not applicable.

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
