# Peer review of "Biological Properties of SARS-CoV-2 Variants: Epidemiological Impact and Clinical Consequences"

_vaccines, 2022, doi:10.3390/vaccines10060919_

Round 1

Reviewer 1 Report

This is a comprehensive review of SARS CoV-2 variants and they characteristics.  The style is fairly anecdotal, referencing many studies.  in some cases, they don't agree.  eg.  line 278 said Ro in USA was 1.56, while line 284 said it was 3.87.  These studies appeared to be over the same early time period.  

  1.  I think this would. be improved if it could be shortened and the information summarized in more tables or graphs.  
  2. Line 61 talks about mutations per cycle, while later in that section, they speak of mutations per year.  I assume cycle meant replication cycle but it's hard to compare with years
  3. Lin3 394:  Please explain SGTF better and what caused it.
  4. It would be interesting to know whether long COVID was affected differently by these variants.
  5. Also, are there human genetic variants that correlate with hospitalization or its lack, etc.?

Author Response

Reviewer 1

This is a comprehensive review of SARS CoV-2 variants and they characteristics.  The style is fairly anecdotal, referencing many studies.  in some cases, they don't agree.  eg.  line 278 said Ro in USA was 1.56, while line 284 said it was 3.87.  These studies appeared to be over the same early time period.  

  1.  I think this would. be improved if it could be shortened and the information summarized in more tables or graphs.  

Thank you for your comment. We intended to make the review as comprehensive as possible. Figures 1 and 2 as well as table A1 provide a summary of the entire review study.

  1. Line 61 talks about mutations per cycle, while later in that section, they speak of mutations per year.  I assume cycle meant replication cycle but it's hard to compare with years

Thank you for highlighting this. Please note that the mutations per cycle have been replaced with mutations per year, and the values have been updated to reflect the literature.

  1. Lin3 394:  Please explain SGTF better and what caused it.

Please note that the definition of SGTF has been clarified.

  1. It would be interesting to know whether long COVID was affected differently by these variants.

This information has been updated in the manuscript. However, more research is needed to address all variants.

  1. Also, are there human genetic variants that correlate with hospitalization or its lack, etc.?

Thank you for your comment. According to research, human genetics variants affect the severity of the virus. The manuscript has been updated to include this information, noting that the genetic studies are done irrespective of circulating variants. 

Reviewer 2 Report

In the manuscript "SARS-CoV-2 Variants Biological Properties: Epidemiological Impact and Clinical Consequences",  Hoteit and Yassine systemically reviewed the epidemiological impact, clinical consequence, and protection strategies in the world. The manuscript is well written, and provide overall information of the SARS-CoV-2 variants. The paper could be accepted after minor revision:

(1) In figure 2, the information of Omicron should be included, which is of significance for detection and prevenntion of the prevalent of SARS-CoV-2.

(2) The authors mainly reviewed the frontiers of SARS-CoV-2 mRNA vaccines. The applicaiton and effectiveness of non-mRNA vaccines, such as the inactivated vaccinces produced by China should also be introduced.

(3) Besides vaccines, clinical strategies to  treat the COVID-19 disease should also be included in this paper.

Author Response

In the manuscript "SARS-CoV-2 Variants Biological Properties: Epidemiological Impact and Clinical Consequences", Hoteit and Yassine systemically reviewed the epidemiological impact, clinical consequence, and protection strategies in the world. The manuscript is well written, and provide overall information of the SARS-CoV-2 variants. The paper could be accepted after minor revision:

  • In figure 2, the information of Omicron should be included, which is of significance for detection and prevenntion of the prevalent of SARS-CoV-2.

Thank you for your comment. Please note that we have updated figure 2 to include Omicron (1-5).

  • The authors mainly reviewed the frontiers of SARS-CoV-2 mRNA vaccines. The applicaiton and effectiveness of non-mRNA vaccines, such as the inactivated vaccinces produced by China should also be introduced.

According to several research studies, inactivated vaccinations are less effective than other vaccines (mRNA in particular). Hence, we focused on the most effective vaccines in this review, noting that the effectiveness of the inactivated vaccine is mentioned in section 2.4.6 page 33.

  • Besides vaccines, clinical strategies to treat the COVID-19 disease should also be included in this paper.

The manuscript has been updated to include clinical treatment options for COVID 19.

This manuscript is a resubmission of an earlier submission. The following is a list of the peer review reports and author responses from that submission.